EMBO
Molecular Medicine

# LAG3 is not expressed in human and murine neurons and does not modulate α-synucleinopathies

Marc Emmenegger[1],[†] , Elena De Cecco[1],[†] , Marian Hruska-Plochan[2],[†] , Timo Eninger[3,4],
Matthias M Schneider[5] , Melanie Barth[3,4], Elena Tantardini[2] , Pierre de Rossi[2],
Mehtap Bacioglu[3,4] , Rebekah G Langston[6], Alice Kaganovich[6], Nora Bengoa-Vergniory[7] ,
Andrès Gonzalez-Guerra[1], Merve Avar[1] , Daniel Heinzer[1] , Regina Reimann[1], Lisa M Häsler[3,4],
Therese W Herling[5], Naunehal S Matharu[5], Natalie Landeck[6] , Kelvin Luk[8] , Ronald Melki[9] ,
Philipp J Kahle[3,10], Simone Hornemann[1] , Tuomas P J Knowles[5,11], Mark R Cookson[6] ,
Magdalini Polymenidou[2], Mathias Jucker[3,4] & Adriano Aguzzi[1],*

## Abstract

While the initial pathology of Parkinson's disease and other α-synucleinopathies is often confined to circumscribed brain regions, it can spread and progressively affect adjacent and distant brain locales. This process may be controlled by cellular receptors of α-synuclein fibrils, one of which was proposed to be the LAG3 immune checkpoint molecule. Here, we analysed the expression pattern of LAG3 in human and mouse brains. Using a variety of methods and model systems, we found no evidence for LAG3 expression by neurons. While we confirmed that LAG3 interacts with α-synuclein fibrils, the specificity of this interaction appears limited. Moreover, overexpression of LAG3 in cultured human neural cells did not cause any worsening of α-synuclein pathology *ex vivo*. The overall survival of A53T α-synuclein transgenic mice was unaffected by LAG3 depletion, and the seeded induction of α-synuclein lesions in hippocampal slice cultures was unaffected by LAG3 knockout. These data suggest that the proposed role of LAG3 in the spreading of α-synucleinopathies is not universally valid.

**Keywords** LAG3; neurodegeneration; prionoids; α-synuclein
**Subject Category** Neuroscience

## Introduction

Lymphocyte-activation gene 3 (LAG3) is an inhibitory immune checkpoint molecule. It may represent a therapeutic target against solid and haematologic tumours (Nguyen & Ohashi, 2015; Andrews *et al*, 2017; Ascierto *et al*, 2017; Lichtenegger *et al*, 2018; Lim *et al*, 2020; Rohatgi *et al*, 2020). LAG3 is expressed in activated T cells, natural killer cells and dendritic cells (Triebel *et al*, 1990; Huard *et al*, 1994, 1997; Hannier & Triebel, 1999; Workman *et al*, 2002, 2009; Maçon-Lemaître & Triebel, 2005; Camisaschi *et al*, 2010), and these findings support its role in the immune system. More recently, it was proposed that LAG3 may function in the central nervous system (CNS) as a receptor of pathogenic α-synuclein assemblies, which are causally involved in Parkinson's disease (PD). Mice devoid of LAG3 were reported to develop lower levels of phosphorylated α-synuclein than wild-type mice upon inoculation with α-synuclein pre-formed fibrils (PFFs). Furthermore, treatment with

1    Institute of Neuropathology, University of Zurich, Zurich, Switzerland
2    Department of Quantitative Biomedicine, University of Zurich, Zurich, Switzerland
3    German Center for Neurodegenerative Diseases (DZNE), Tübingen, Germany
4    Department of Cellular Neurology, Hertie Institute for Clinical Brain Research, University of Tübingen, Tübingen, Germany
5    Yusuf Hamied Department of Chemistry, Centre for Misfolding Diseases, University of Cambridge, Cambridge, UK
6    Cell Biology and Gene Expression Section, Laboratory of Neurogenetics, National Institute on Aging, National Institutes of Health, Bethesda, MD, USA
7    Department of Physiology, Anatomy and Genetics, Oxford Parkinson's Disease Center (OPDC), Oxford University, Oxford, UK
8    Department of Pathology and Laboratory Medicine and Center for Neurodegenerative Disease Research, University of Pennsylvania Perelman School of Medicine, Philadelphia, PA, USA
9    Laboratory of Neurodegenerative Diseases, CNRS, Institut François Jacob (MIRCen), CEA, Fontenay-aux-Roses, France
10   Department of Neurodegeneration, Hertie Institute for Clinical Brain Research, University of Tübingen, Tübingen, Germany
11   Cavendish Laboratory, Department of Physics, University of Cambridge, Cambridge, UK
     *Corresponding author. Tel: +41 44 255 21 07; E-mail: adriano.aguzzi@usz.ch
     †These authors contributed equally to this work

anti-LAG3 antibodies attenuated the spread of pathological α-synuclein and drastically lowered the aggregation *in vitro* (Mao *et al*, 2016).

This finding, if confirmed, could have far-reaching implications. PD is a common neurodegenerative movement disorder (Beitz, 2014; Deweerdt, 2016; Jankovic, 2017) that causes a high level of suffering to the affected patients and their families. Histologically, PD is characterized by α-synuclein inclusions known as Lewy bodies whose accumulation is associated with neurodegeneration (Dickson, 2012; Mullin & Schapira, 2015; Corbillé *et al*, 2016). These inclusions affect the Substantia nigra and other mesencephalic regions as well as, in some cases, the amygdala and neocortex (Dickson, 2018). Growing evidence suggests that α-synuclein aggregates spread from cell to cell (Volpicelli-Daley *et al*, 2011; Volpicelli-Daley *et al*, 2014), by a "prionoid" process of templated conversion (Aguzzi, 2009; Jucker & Walker, 2018; Kara *et al*, 2018; Henderson *et al*, 2019; Karpowicz *et al*, 2019; Uemura *et al*, 2020; Kara *et al*, 2021). It is thought that interrupting transmission of α-synuclein may slow down or abrogate the disease course. Although a more general pathogenic role for LAG3 in the progression of prion disorders has been ruled out (Liu *et al*, 2018), impairing the binding of α-synuclein fibrils to neuronal LAG3 may still constitute an attractive target for small drugs or immunotherapy of PD.

Here, we have analysed LAG3 expression and α-synuclein binding in mouse and human model systems. Additionally, we studied the propagation of pre-formed fibrils (PFFs) of α-synuclein in neural stem cell (NSC)-derived neural cultures in the presence or absence of LAG3. Finally, we have investigated the impact of LAG3 on survival in ASYN$^{A53T}$ transgenic mice (a model of Parkinson's disease) expressing wild-type LAG3 as well as hemizygous or homozygous deletions thereof. We failed to detect neuronal expression of LAG3 and were unable to establish any role for LAG3 in α-synucleinopathies *in vitro* and *in vivo*.

# Results

## Absence of endogenous LAG3 from neuronal cell lines, NSC-derived neural cultures and human brain samples

The sequence homology between human and mouse LAG3 proteins is < 70%. This may limit the cross-species reactivity of anti-LAG3 antibodies. We therefore asked whether available antibodies bind the extracellular domain of human or mouse LAG3. We coated 384-well microplates with recombinant human LAG3$_{23-450}$ or mouse LAG3$_{24-442}$ and measured binding of 8 commercially available anti-LAG3 antibodies by enzyme-linked immunosorbent assay (ELISA) (Fig 1A). All antibodies except LSB15026 bound exclusively human or mouse LAG3, but not both. This suggests that many relevant LAG3 epitopes differ between species. Western blot analysis confirmed that the mouse monoclonal antibody 4-10-C9 used in Mao *et al* (2016) did not bind human LAG3 as either recombinant protein or overexpressed by lentivirally transduced murine primary cultures, whereas murine LAG3 was detected (Fig 1B).

We then attempted to identify a human neural cell line that expresses LAG3. We therefore immunoblotted five different human cell lines and included activated T lymphocytes for control. We did not detect bands specific for LAG3 in any of the human glial,

neuronal or control (HEK293T) cell lines (Fig 1C). We then induced differentiation of human neural stem cells (NSC) for which we had available concomitant single-cell RNA sequencing (scRNAseq) data (Hruska-Plochan *et al*, 2021), and immunoblotted their lysates. For control, NSC-derived cultures were additionally transduced with a lentiviral plasmid encoding human LAG3. Western blotting did not reveal any LAG3-specific bands in non-induced cultures (Fig 1D), and scRNAseq yielded only minimal counts for the LAG3 transcript in neurons (N), astrocytes (AST) and mixed glial (MG) cells (Fig 1 E). The counts observed for LAG3 in these cell types were comparable to those of the other T-cell checkpoints, T-cell immunoreceptor with Ig and ITIM domains (TIGIT) and T-cell immunoglobulin and mucin domain-containing protein 3 (TIM3 or HAVCR2) whereas neuronal or astrocytic markers displayed the expected transcriptional profiles (Fig EV1A). We also confirmed the absence of LAG3 signal in immunoblots of dopaminergic neuronal cultures from control lines and PD patients carrying a N370S polymorphism in the glucocerebrosidase (GBA) gene (Fig EV1B and F), further suggesting that LAG3 is not expressed in neurons.

We next investigated whether LAG3 expression varies between individual cells, potentially preventing its detection by bulk methodologies such as Western blotting despite robust expression by rare single cells. We therefore performed immunofluorescence stainings on NSC-derived human neural cultures. Again, no LAG3 could be identified, whereas neural cultures lentivirally transduced with a plasmid encoding LAG3 showed obvious positivity with two different antibodies (Fig 1G).

Subsequently, we investigated human brain areas for LAG3 expression. We selected post-mortem brain samples of Substantia nigra and frontal cortex for immunoblotting. Activated T lymphocytes and lymphoepithelial tissue of tonsils were used as positive controls due to their high expression of LAG3. Western blotting failed to reveal the presence of LAG3 in human brain samples (Fig 1H).

Finally, we interrogated a single-nucleus (sn) RNAseq human brain dataset that we have recently described (Saez-Atienzar *et al*, 2021) for LAG3 expression across different cell types. These data are derived from 21 dorsolateral prefrontal cortices from 16 neurologically healthy donors (median age 36 years, range: 16–61 years, male:female ratio = 1:1) and clustered and annotated using known gene expression markers as specified (Saez-Atienzar *et al*, 2021). We saw no *LAG3* signals above background for any of 34 identified cell clusters, including 13 clusters of excitatory and 11 subtypes of inhibitory neurons, oligodendrocytes (ODC), oligodendrocyte precursor cells (OPC), microglia (MGL), astrocytes (AST) and endothelial cells (EC) (Fig 1I). Similar results were obtained by examining available scRNAseq and snRNAseq datasets from juvenile and adult human brain tissue samples including Substantia nigra (Zhang *et al*, 2016; Welch *et al*, 2019; Agarwal *et al*, 2020). These results corroborate the absence of detectable LAG3 expression in human neurons using snRNAseq.

## Absence of endogenous LAG3 in murine brain samples

As the inability to detect LAG3 in human neuronal samples contradicts previous observations (Mao *et al*, 2016), we analysed LAG3 expression in mouse brains. Prior RNAseq data reported that *Lag3* is poorly expressed in both hippocampus and cerebellum of WT mice (Liu *et al*, 2018) and likely absent in murine neurons (Zhang *et al*,

2014). Western blot of total brain homogenates from C57BL/6J wild-type mice did not show any identifiable LAG3-specific band (Fig 2A). Incubation with a different primary anti-LAG3 antibody (Fig EV1C) and enrichment by immunoprecipitation (Fig EV2A) confirmed the lack of LAG3 in wild-type mouse brain homogenates.

Next, we prepared mixed neuroglial and glial cortical cultures from C57BL/6J mice and assessed LAG3 protein expression. The anti-mitotic compound AraC was added to the mixed cortical cultures to enrich for neurons and eliminate glia (Fig 2B, lower panels). Western blotting (Fig 2B) did not reveal any endogenous

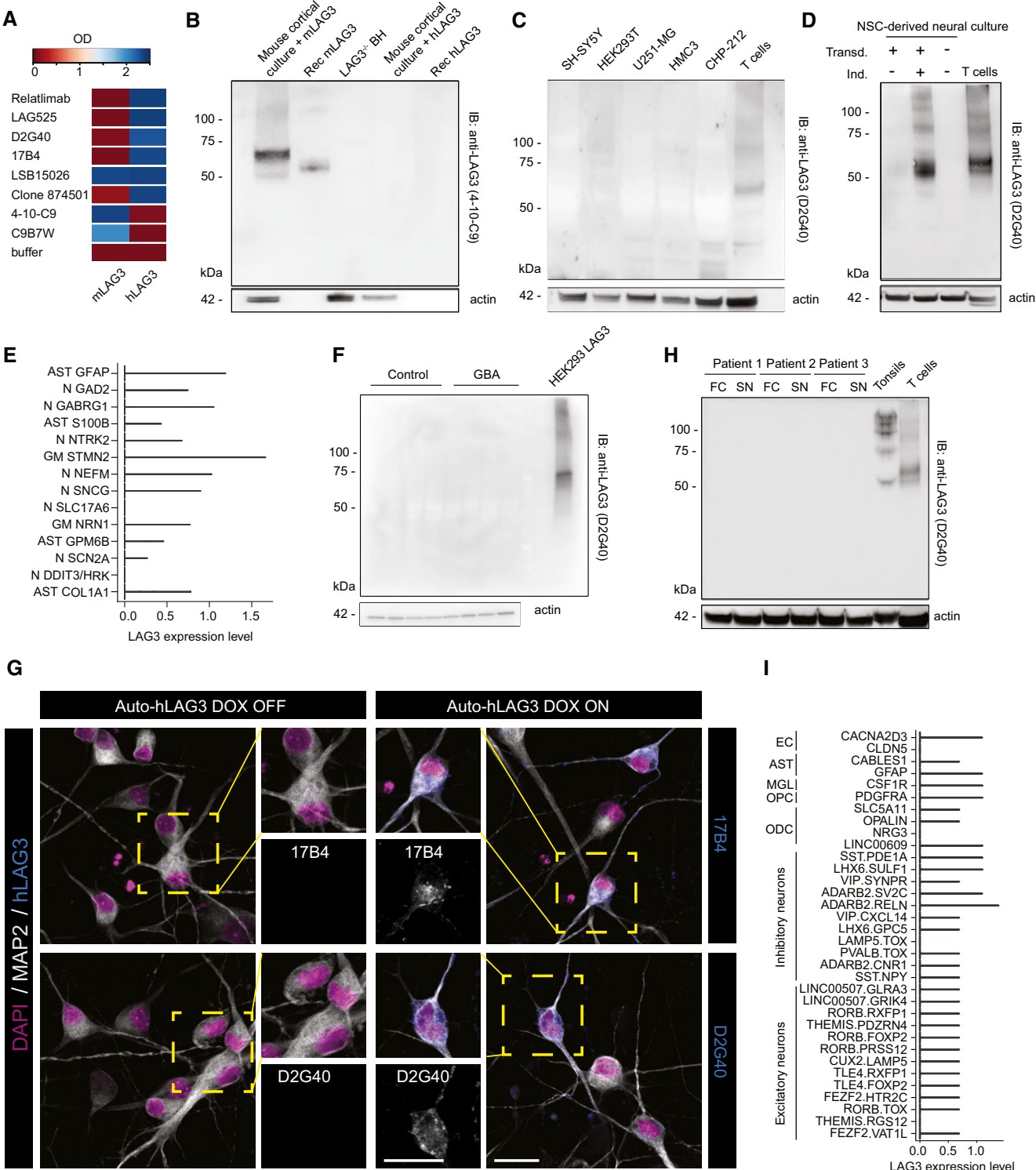

**Figure 1.**

**Figure 1. Absence of expression of LAG3 in human brain cells.**

A   Binding of eight commercial antibodies to recombinant human LAG3$_{23-450}$ and murine LAG3$_{24-442}$ via indirect ELISA. Seven out of eight antibodies bound either human or mouse LAG3, while one antibody (LSB15026) recognized both species.

B   Specific detection of murine but not human LAG3 using 4-10-C9 anti-LAG3 antibody is confirmed with Western blotting.

C   No detection of human LAG3 in neuronal or glial cell lines of human origin. The band for LAG3 was detected in activated T cells.

D   No band for human LAG3 could be detected with Western blot in lysates of fully differentiated human NSC-derived neural cultures.

E   Violin plot showing the RNA expression levels of human LAG3 in human NSC-derived neural cultures. Identities annotate different clusters: Neuronal clusters are comprised of the following markers: GAD2, GABRG1, NTRK2, NEFM, SNCG, SLC17A6, SCN2A, DDIT3/HRK. Mixed glial clusters are defined by the following markers: GFAP, S100B, STMN2, NRN1, GPM6B, COL1A1, with astrocyte-specific clusters characterized by GFAP, S100B, GPM6B, COL1A1. LAG3 cannot be evidenced in any of the clusters beyond few random events. Data shown from 5,476 unique analysed cells from one out of two independent biological replicates.

F   Dopaminergic neuronal cultures from control lines and glucocerebrosidase (GBA) N370S PD patients were immunoblotted for the presence of LAG3. No band for LAG3 could be observed in neurons.

G   Using high power, high-resolution laser scanning confocal microscopy, no human LAG3 signal could be detected in human neurons (Auto-hLAG3 transduced, DOX OFF) by two different anti-human LAG3 antibodies (17B4 and D2G40; left panel and zoomed-in insets) whereas LAG3 was clearly detected in human neurons induced to express hLAG3 (DOX ON; right panel and zoomed-in insets). Scale bars 25 μm.

H   Human brain homogenates from autopsy material were immunoblotted for the presence of LAG3. No band for LAG3 could be evidenced in any of the brain homogenates. Control samples (tonsils and activated human T cells) show expected bands.

I   Nuclei were isolated from 21 human dorsolateral prefrontal cortices from 16 donors, isolated and were subjected to snRNAseq. Expression levels were quantified and are shown as violin plots. The LAG3 transcripts are non-detectable in all 34 distinct cell types, including multiple excitatory and inhibitory neurons, oligodendrocytes (ODC), oligodendrocyte precursor cell (OPC) microglia (MGL), astrocytes (AST) or endothelial cells (EC). Cluster markers as detailed in Saez-Atienzar et al, 2021.

Source data are available online for this figure.

LAG3 in any of the samples, even after enrichment by immunoprecipitation with anti-mouse LAG3 antibody 4-10-C9 from 500 μg of total protein (Fig EV2B). Using a LAG3 sandwich ELISA, we assessed LAG3 expression in primary cultures, murine synaptoneurosome preparations and brain region-specific preparations. We did not find LAG3 expression in any of the samples, in contrast to activated T cells used for control (Fig 2C). Immunofluorescence staining of neuronally enriched primary cultures led to the same results, with LAG3 being detectable only in the lentivirally transduced cultures that were used as positive controls (Fig 2D).

We next quantified *Lag3* mRNA from mixed neuroglial and glial cortical cultures by RT–qPCR (Fig 2E). By interpolation of the threshold cycle ($C_T$) of each sample into a standard curve, we estimated the presence of around 1–2 transcripts/ng of RNA in mixed cortical cultures and neuron-enriched cultures, whereas in mixed glial cultures, *Lag3* mRNA was slightly more abundant (4–5 transcripts/ng of RNA). These values are close to what we observed for a LAG3$^{-/-}$ mouse brain homogenate (1–2 transcripts/ng of RNA) and indicate that these *Lag3* signals were nonspecific. In contrast, around 1,600 transcripts/ng of RNA were counted in activated T cells.

To further expand our analyses, we performed scRNAseq of microdissected ventral midbrain and striatum of 1-year-old mice that had been given a single striatal injection of PBS or LPS ($n = 9$ and 8, respectively) as previously described (Russo et al, 2019). We then assigned 76,305 cells to major cluster using markers (Fig EV2C). We specifically investigated transcript levels of murine *Lag3* in endothelial cells, several macrophage subtypes (MPH), astrocyte subtypes, red blood cells (RBC), vascular smooth muscle cells (VSMC), neural precursor cells (NPC), microglia subtypes (MGL), oligodendrocyte subtypes (ODC), T cells, neuron subtypes, oligodendrocyte precursor cells (OPC), ependymal cells (Epnd), choroid plexus cells (ChP) and fibroblasts (Fibr) (Fig EV2D). Of all identified murine cells, only transmembrane protein 119 (TMEM119)-positive microglia showed modest expression of *Lag3*, marginally above background (Fig 2F). Of interest, the activation of microglia with lipopolysaccharide (LPS) led to decreased expression of *Lag3* in microglia (Fig EV2E). However, and consistent with the negative results for detection of LAG3 in human snRNAseq, we did not detect murine *Lag3* in neurons or astrocytes under any condition examined.

## Promiscuous binding of α-synuclein fibrils questions selectivity towards LAG3

The above observations suggest that the neuroprotective effects exerted by antibodies to residues 52–109 of the LAG3 D1 domain, or the depletion of LAG3 (Mao et al, 2016), may not be required for the interaction between neuronal LAG3 and α-synuclein fibrils. However, α-synuclein fibrils binding to LAG3 could still be an important component involved in PD by an unknown mechanism. To investigate the interaction between LAG3 and α-synuclein fibrils, we performed ELISA by coating human LAG3$_{23-450}$, the human LAG3 structural homologue, CD4 (Triebel et al, 1990; Bae et al, 2014), the microtubule-binding domain (MTBD) of human tau protein, the prion protein PrP$^C$, apolipoprotein E3 (APOE3), TAR DNA-binding protein 43 (TDP-43), bovine serum albumin (BSA) and arachis hypogaea 2 (Ara h 2), a major peanut allergen, to individual wells of a microtiter plate. We then incubated ELISA plates with serial dilutions (concentration of monomer equivalent: 125 nM–30 pM) of α-synuclein fibrils and filaments of apoptosis-associated speck-like protein containing a CARD (ASC). If binding occurs, the fibrils would be retained by the antigen and would be detected by antibodies. ASC filaments yielded the expected dose–response curves when directly coated onto microplates (Fig EV3) but did not bind any of the proteins presented (Fig 3A). While α-synuclein fibrils (but not monomeric α-synuclein) interacted with LAG3, they exhibited similar binding to MTBD of human tau protein and to CD4 (Fig 3B). PrP$^C$, the APOE3 as well as TDP-43, was found to interact less strongly, whereas BSA, Ara h 2 and uncoated conditions did not display any binding in our system.

The observations detailed above suggest a promiscuous binding of α-synuclein fibrils to many proteins including LAG3, MTBD and CD4, suggestive of surface effects leading to nonspecific interactions. We therefore employed a microfluidic-based technology (Arosio et al, 2016; Scheidt et al, 2019; preprint: Schneider et al, 2020)

to characterize the receptor-ligand interaction. Microfluidic diffusional sizing (MDS) measures the hydrodynamic radius ($R_h$) of a fluorescently labelled protein and can characterize binding by displaying an increase in $R_h$ upon ligand binding. Increasing concentrations of α-synuclein fibrils did not induce a change in the hydrodynamic radius of fluorescently labelled LAG3$_{23–434}$, CD4 or BSA

(Fig 3C), whereas a labelled antibody directed against α-synuclein (MJFR1) triggered the expected concentration-dependent size increase. A progressive increase in the concentration of monomeric α-synuclein did not lead to a rise in the hydrodynamic radius of labelled LAG3; however, both anti-LAG3 antibody (Relatlimab) and FGL1, a recently discovered interaction partner of LAG3 (Wang *et*

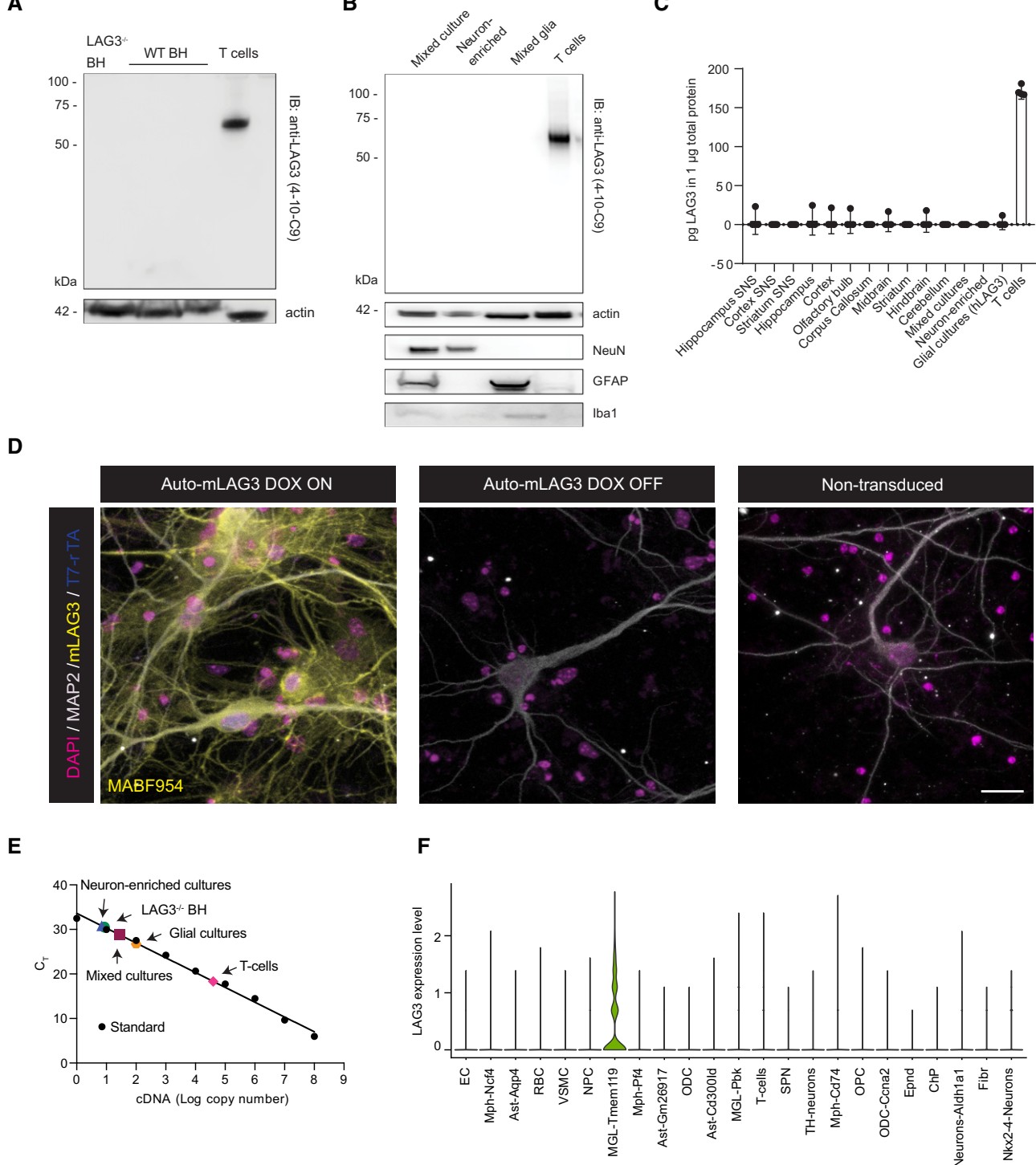

**Figure 2.**

**Figure 2. Absence of expression of LAG3 in mouse brain cells, particularly in neurons.**

A  Brain homogenates from LAG3 KO and WT mice were immunoblotted for the presence of murine LAG3. Western blot does not show a specific band other than for activated murine T cells.

B  Mixed neuroglial, neuron-enriched and glial cortical cultures were assayed for the presence of LAG3 where no band could be found. Activated T cells display band at expected size.

C  Lysates of synaptoneurosomes (SNS) of different brain regions and mouse brain region-specific areas as well as murine activated T cells were assayed for the presence of LAG3 with sandwich ELISA. No evidence for LAG3 could be found in any of the samples other than in activated T cells. Expression is shown in pg LAG3 per µg total protein as mean values of technical quadruplicates with 95% confidence intervals.

D  Using high power, high-resolution laser scanning confocal microscopy, no mouse LAG3 signal could be detected in murine primary neuronal cultures (Auto-mLAG3 transduced, DOX OFF and non-transduced neurons) using anti-mouse LAG3 antibody (MABF954) whereas LAG3 was clearly detected in primary neurons induced to express mLAG3 (Auto-mLAG3 transduced, DOX ON). Scale bars 20 µm.

E  RT–qPCR-derived $C_T$ values to measure the cDNA copy number. Mixed neuroglial cultures, glial cultures and neuron-enriched cultures show transcript levels that are close to indistinguishable to those from LAG3$^{-/-}$ BH. Conversely, around 1,600 transcripts/ng of RNA were counted in activated T cells.

F  Murine mesencephalon and striatum of 1-year-old mice ($n = 17$ in total) were interrogated with scRNAseq. Violin plot showing expression values of Lag3, in 76,305 assigned cells, between identified cell clusters, mainly expressed in TMEM119-positive microglia cells. Cluster markers are detailed in Russo et al, 2019.

Source data are available online for this figure.

al, 2019), displayed clear binding (Fig 3D), with affinities of $4.82 \pm 0.71$ nM and $232 \pm 79$ nM, respectively. Given that no significant binding was detected, we conclude that the affinity of LAG3 for α-synuclein fibrils, if any, is micromolar or less.

To shed light onto the interaction between α-synuclein fibrils and LAG3, we performed co-immunoprecipitation experiments. SH-SY5Y cells overexpressing hLAG3 were incubated with monomeric or fibrillary α-synuclein in live cells for 3 h at 37°C to allow binding at the cell membrane. Then, immunoprecipitations with antibodies to LAG3 or α-synuclein were performed. Immobilized LAG3 co-precipitated α-synuclein fibrils, whereas co-precipitation with both monomeric and fibrillar α-synuclein resulted in only a faint band for LAG3 (Fig 3E and F). A major portion of both LAG3 and α-synuclein fibrils remained in the unprecipitated soluble fraction.

## Membrane-bound LAG3 does not affect the uptake and seeding of α-synuclein PFFs in human neural cultures

The collective biochemical data presented above led us to question whether there is a physiologically relevant interaction between α-synuclein fibrils and LAG3. We therefore tested its functional relevance in a human cell-based system. We differentiated NSCs into human neural cultures previously shown to be devoid of LAG3 (Fig 1D and E) and transduced them with a lentivirus encoding inducible expression of human LAG3. These cultures expressed α-synuclein (Fig EV4A). We then added sonicated α-synuclein-pre-formed fibrils (PFFs) (Volpicelli-Daley et al, 2014) and incubated the cells with them for 14–28 days. Finally, we stained the phosphorylated form (pS129) of α-synuclein with two different phosphorylated α-synuclein-specific antibodies. We then compared the amount of α-synuclein fibrils uptake and seeding through intracellular pS129 α-synuclein accumulation, in cultures that did not express (Fig EV4B) or expressed (Fig EV4C) LAG3. We found no difference in the proportion of cells containing pS129 α-synuclein between the two conditions using the 81A (Fig 4A) or EPY1536Y anti-pS129 α-synuclein antibodies (Fig 4B). Quantification of images using a trained ilastik-based algorithm for pixel segmentation failed to show any effects of LAG3 expression (Fig 4C and D). This suggests no significant role for neuronal LAG3 in the uptake and, therefore, in the downstream processes, including the transmission, of α-synuclein fibrils in this paradigm.

## Similar survival of ASYN$^{A53T}$ LAG3$^{-/-}$, LAG3$^{+/-}$ or LAG3$^{+/+}$ mice

Human LAG3 may be functionally dissimilar to its murine counterpart. Hence, the negative data shown above do not negate an effect of Lag3 in mice. We therefore examined the impact of LAG3 expression on the survival of mice expressing human α-synuclein$^{A53T}$ under the transcriptional control of the mouse Thy1.2 promoter (Van Der Putten et al, 2000). Since the expression of endogenous α-synuclein is a key requirement for its propagation (Volpicelli-Daley et al, 2014; Polinski et al, 2018), and since expression levels of α-synuclein influence the propensity of aggregate formation (Hayashita-Kinoh et al, 2006), we first assessed whether α-synuclein expression is similar in ASYN$^{A53T}$-expressing LAG3$^{-/-}$ and LAG3$^{+/+}$ mice. No differences were found in cerebrospinal fluid (CSF) collected from aged mice with Quanterix Single Molecule Arrays (SIMOA) (Fig EV5), indicating that LAG3 does not modulate the levels of the transgene.

We then bred ASYN$^{A53T}$;LAG3$^{-/-}$, ASYN$^{A53T}$;LAG3$^{+/-}$, ASYN$^{A53T}$; LAG3$^{+/+}$ and nontransgenic LAG3$^{-/-}$ mice ($n = 8$, 9, 13 and 8, respectively, including males and females). All mice expressing ASYN$^{A53T}$, but none of the nontransgenic LAG3$^{-/-}$ mice, developed severe α-synucleinopathy and were sacrificed with a median survival of A53T α-synuclein mice of 263 days regardless of LAG3 genotype (95% CI: 240–277; Fig 5A). Consistent with these results, LAG3 knockout did not induce histologically detectable changes in the mesencephalic distribution patterns of pS129 α-synuclein or thioflavin S-positive aggregates (Fig 5B) of end-stage symptomatic mice. These results suggest that LAG3 plays little, if any, role in the propagation of α-synuclein pathology in vivo.

## No difference in pS129 α-synuclein in hippocampal organotypic slice cultures of A53T LAG3$^{-/-}$ or LAG3$^{+/+}$ mice

The effect of LAG3 in mice may be contingent on the utilization of α-synuclein PFFs as a seed. Alternatively, LAG3 may influence early α-synuclein lesion formation without modifying the survival or the end-stage pathology of mice. To this end, we cultured hippocampal slices of A53T α-synuclein TG or α-synuclein WT mice KO or WT for LAG3, respectively, and inoculated them with 5 µg α-synuclein PFFs. Organotypic slices were kept in culture for 5 weeks, and the pS129- or thioflavin S-positive area was assessed (Barth et al, 2021).

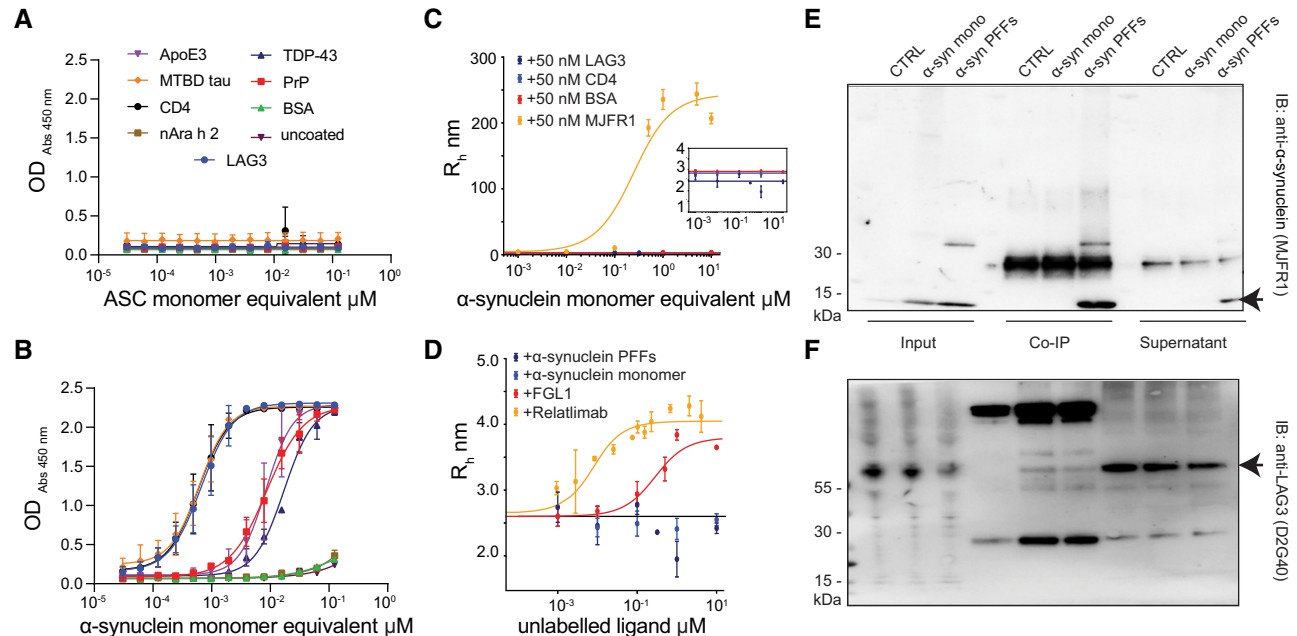

**Figure 3. Characterization of binding properties between α-synuclein PFFs and LAG3 reveals promiscuous character of α-synuclein.**

A, B   Binding of ASC filaments (A) or α-synuclein PFFs (B) to multiple proteins. Serial dilutions of ASC filaments and α-synuclein PFFs were performed, and bound fractions were detected using anti-ASC or anti-α-synuclein (MJFR1) antibodies. ASC filaments did not interact with any of the proteins (A). Conversely, α-synuclein PFFs strongly bound LAG3, CD4 and MTBD tau and showed moderate binding properties to PrP[C], APOE3 and TDP-43. No binding between α-synuclein PFFs and nAra h 2 or BSA could be detected. Values represent means ± SD of technical duplicates.

C, D   Diffusional sizing of α-synuclein PFFs with fluorescently conjugated LAG3, CD4, BSA and MJFR1 α-synuclein antibody (C) and of fluorescently labelled LAG3 with monomeric α-synuclein, α-synuclein PFFs, FGL1 and Relatlimab anti-LAG3 antibody (D). An increase in the hydrodynamic radius ($R_h$) of α-synuclein PFFs is seen with increasing concentrations for MJFR1 but not for LAG3 or any of the control proteins, also not at a lower scale (see insert) (C). Similarly, the hydrodynamic radius of labelled LAG3 increases with higher concentrations of Relatlimab and FGL1 but does not change with α-synuclein PFFs or monomeric α-synuclein (D). Values are given as means ± SD of technical triplicates.

E, F   Co-immunoprecipitation of LAG3 and subsequent immunoblotting with anti-α-synuclein antibody (MJFR1) (E) and co-immunoprecipitation of α-synuclein (monomeric or PFFs) and subsequent immunoblotting with anti-LAG3 antibody (D2G40) (F). Using the anti-α-synuclein antibody for pulldown resulted in LAG3 bands with both monomeric as well as fibrillar α-synuclein. The usage of anti-LAG3 for pulldown led to the specific detection of α-synuclein PFFs, although the PFF fraction predominates also in the input and supernatant. Bands for α-synuclein (E) and LAG3 (F) are indicated by arrows.

Source data are available online for this figure.

Non-seeded organotypic slices did not show any sign of pS129 α-synuclein (Fig 5C) or thioflavin S positivity (Fig 5D), while seeded α-synuclein WT slices showed moderate pS129 α-synuclein- or thioflavin S-positive areas and A53T slices had more prominent pathology, as previously reported (Barth *et al*, 2021). However, no significant difference was found between LAG3 KO and WT in slices derived from A53T transgenic or wild-type mice, indicating that LAG3 does not play an essential role in the uptake and propagation of α-synuclein PFFs in this system.

## Discussion

LAG3 was named a selective receptor for pathogenic α-synuclein assemblies using an exhaustive series of *in vitro* and *in vivo* experiments with a variety of genetic and pharmacological tools (Mao *et al*, 2016). Yet, there is little evidence for a role of LAG3 in human α-synucleinopathies (Liu *et al*, 2018; Cui *et al*, 2019; Guo *et al*, 2019). This situation prompted us to revisit the interaction between LAG3 and α-synuclein conformers. When assessing cell lines,

NSC-derived neural cultures or organ homogenates for the presence of human or murine LAG3, we selected antibodies appropriate to the species investigated and, whenever possible, have used multiple antibodies to increase the confidence in our results. We were unable to detect LAG3 in any of the neuronal samples tested. Thus, we felt the need to investigate further using a broader array of techniques: Western blotting, sandwich ELISA, immunofluorescence, RT–qPCR, snRNAseq and scRNAseq. None of these methods were able to detect endogenous LAG3 expression in human neurons. The most parsimonious interpretation of these collective data is that human neurons do not express LAG3 at appreciable levels. One notable exception is our scRNAseq dataset of mouse ventral midbrain detecting low expression of LAG3 in microglia, in accordance with previous reports (Tasic *et al*, 2016; Galatro *et al*, 2017).

It is unclear whether the lack of detection of LAG3 in human microglia relates to a crucial species difference or to technical limitations of snRNAseq versus scRNAseq, which may be particularly important for microglia (Thrupp *et al*, 2020). Either way, these data and other available data sets (Zhang *et al*, 2014, 2016; Welch *et al*, 2019; Agarwal *et al*, 2020; Almanzar *et al*, 2020) do not support

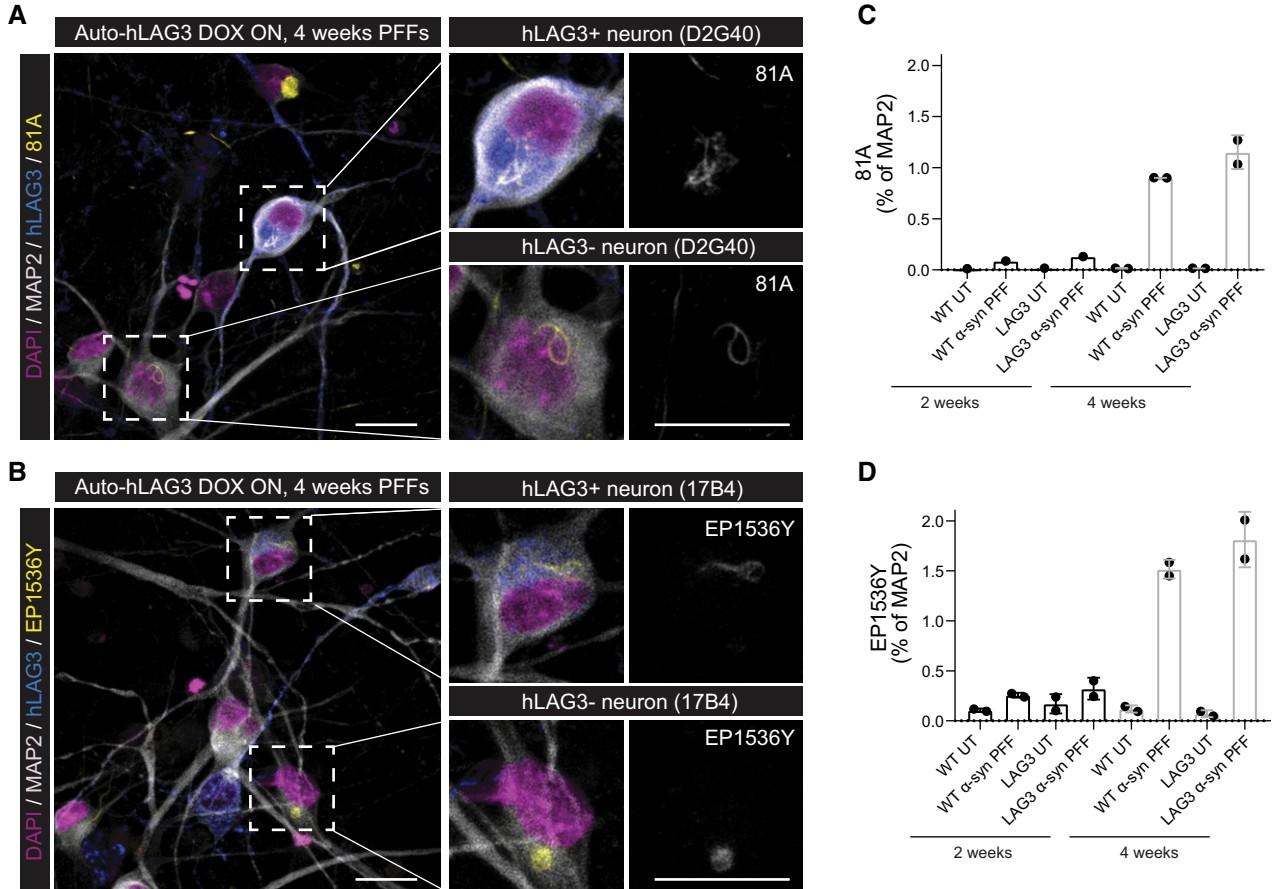

**Figure 4. Propagation of α-synuclein PFFs *in vitro* is not dependent on LAG3 in human NSC-derived neural cultures.**

Human neural cultures transduced by Auto-hLAG3 were treated by α-synuclein PFFs 4 days post-LAG3 expression induction by DOX and kept in culture for 2 or 4 weeks. Both transgenic (hLAG3 D2G40-positive in (A) and 17B4-positive in (B)) and non-transduced, wild-type neurons (selected neurons in zoomed-in insets) propagated α-synuclein and developed characteristic pS129-positive (81A-positive in (A) and EP1536Y-positive in (B)) α-synuclein aggregates. Scale bars 25 μm. Trained ilastik algorithms were used to segment pixels of 81A, EP1536Y and MAP2 stainings imaged by high-content widefield microscope, which were used to quantify the signal of 81A-positive (C) and EP1536Y-positive (D) α-synuclein aggregates expressed as % of MAP2-positive area. Almost the entire wells (182 fields per well) were imaged for every condition and replicate, and each datapoint in the plot represents the entire well. Error bars indicate mean ± SD of biological replicates (duplicates). For few conditions, unicates were used, shown as one dot. One-way ANOVA followed by Tukey's multiple comparison test demonstrated that neurons that did not express LAG3 (DOX OFF) showed no difference in α-synuclein propagation when compared to LAG3-expressing neurons (DOX ON) as demonstrated by two different pS129 α-synuclein antibodies ($P = 0.9998$ for 81A at 2 weeks and $P = 0.2522$ at 4 weeks; $P = 0.9986$ for EP1536Y at 2 weeks and $P = 0.3042$ at 4 weeks).

Source data are available online for this figure.

expression of neuronal LAG3, at least using current technology for single-cell analysis. Thereby, we question one of the foundations of the hypothesis proposed by Mao *et al*, (2016) and discussed elsewhere (Jucker & Heikenwalder, 2016; Wood, 2016; Wong & Krainc, 2017).

A possible confounder may be the cross-immunoreactivity of antibodies between murine and human LAG3, which appears to be rare. Within a collection of commercial anti-LAG3 antibodies, only one out of 8 anti-LAG3 antibodies reacted with both human as well as murine LAG3. Mao et al. immunoblotted the human HEK293FT and SH-SY5Y cell lines alongside mouse cortical cultures using an anti-LAG3 antibody (4-10-C9, MABF954) that appears to specifically recognize LAG3 of murine origin (see supplementary figure 5 in Mao *et al*, 2016).

LAG3 could plausibly play a role in the pathogenesis of α-synucleinopathies, e.g. via complex formation with soluble LAG3

(Guo *et al*, 2019) and subsequent endocytosis by hitherto unidentified neuronal receptors or via targeting immune checkpoints of T cells (Baruch *et al*, 2016; Schwartz, 2017; Liu & Aguzzi, 2019) despite not being pulled down upon brief exposure of mouse primary neurons and astrocytes to α-synuclein fibrils (Shrivastava *et al*, 2015). The interaction between LAG3 and fibrillary α-synuclein is therefore of interest. α-Synuclein fibrils, in contrast to apoptosis-associated speck-like protein containing a CARD (ASC), are promiscuous; yet, they bind some proteins much better than others. LAG3, the microtubule-binding domain of human tau and CD4, interacted more than PrP$^C$, APOE3 and TDP-43, whereas BSA and Ara h 2 showed even less interaction. However, the binding of α-synuclein fibrils to CD4 stands in direct opposition to data presented earlier (Mao *et al*, 2016). FGL1, a recently reported interaction partner of LAG3 (Wang *et al*, 2019), could be confirmed to bind LAG3 with microfluidic diffusional sizing (MDS), with an

affinity of 232 ± 79 nM, slightly higher than a previous assessment with Octet bio-layer interferometry analysis suggested (Wang *et al*, 2019). On the other hand, MDS failed to reproduce data obtained by ELISA, as no binding between LAG3 and α-synuclein fibrils occurred, nor with CD4 and α-synuclein fibrils. This could indicate that the nature of interaction differs between LAG3 and α-synuclein

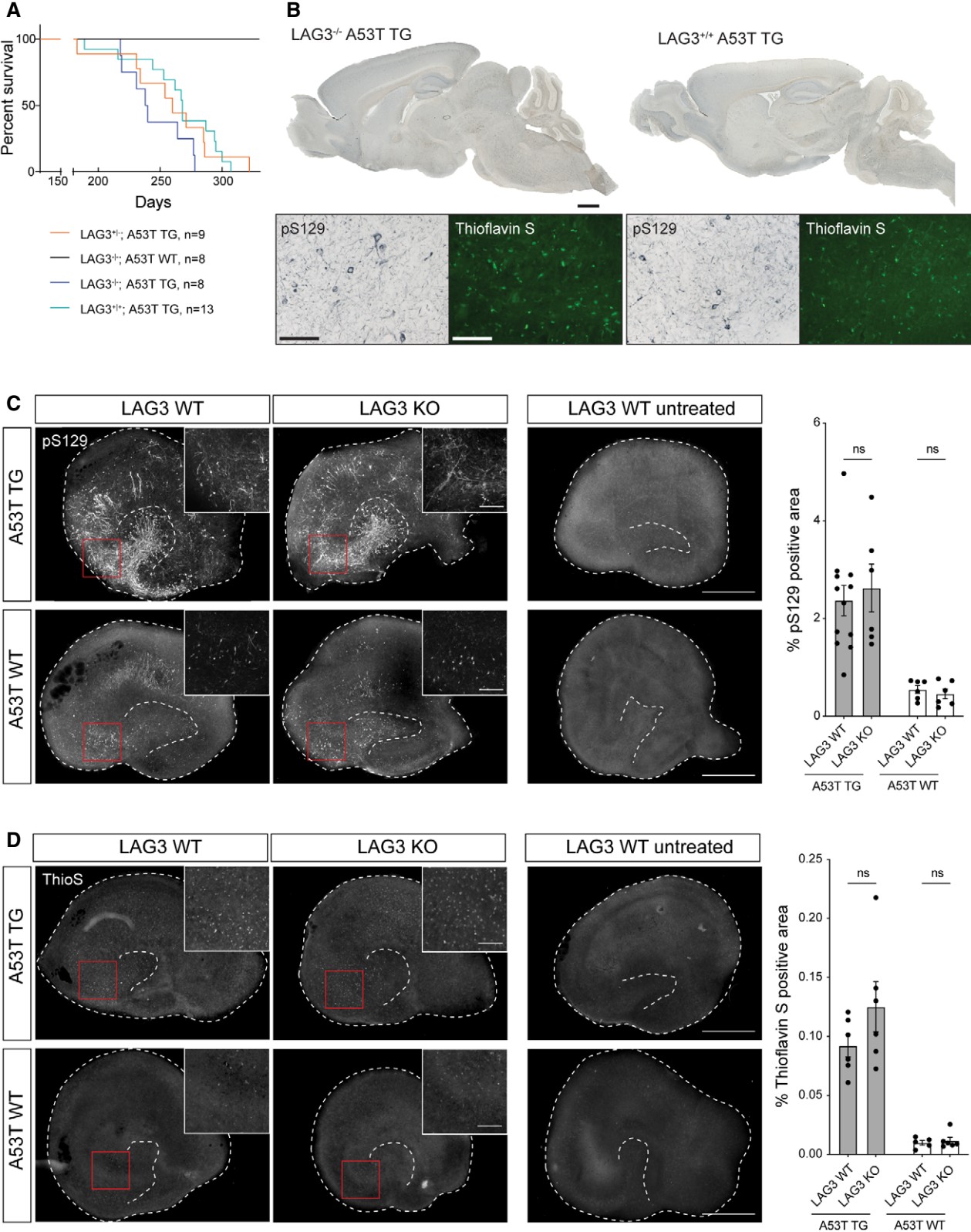

**Figure 5.**

**Figure 5.   Survival in A53T α-synuclein mice and aggregation of α-synuclein *in vivo* and in hippocampal slice cultures is independent from LAG3 expression.**

A   Survival curves of A53T α-synuclein TG mice, with LAG3 knockout, heterozygous LAG3 expression, or LAG3 WT. Median survival was shortest for α-synuclein TG LAG3$^{-/-}$ at 239 days, followed by α-synuclein TG LAG3$^{+/-}$ at 260 days and α-synuclein TG LAG3$^{+/+}$ at 268 days. The survivals of ASYN$^{A53T}$ LAG3$^{-/-}$, LAG3$^{+/-}$ and LAG3$^{+/+}$ mice were similar (Mantel–Cox log-rank test, *P*-value = 0.165). Mice were euthanized and dissected in the late-stage symptomatic phase, which would inevitably result in death within a week. None of the control LAG3$^{-/-}$ α-synuclein WT mice died over the course of the experiment.

B   Immunostaining for hyperphosphorylated α-synuclein aggregates (pS129) and aggregated protein (Thioflavin S) in sections of end-stage symptomatic mice; no obvious difference in terms of staining pattern or number of stained cells was apparent in midbrain or brain stem. Scale bars represent 1 mm (sagittal section) and 100 μm (panel).

C   Immunofluorescence staining and quantification of pS129-positive α-synuclein inclusions in hippocampal slice cultures (HSCs) at 5 weeks post-seeding with 350 μM PFF. Scale bars represent 500 μm (overview), and 100 μm (insets). Shown is mean ± SEM, and individual values are shown; *n* = 6 cultures per group; two-way-ANOVA revealed for LAG3 $F_{(1, 20)}$ = 0.0294; *P* = 0.8656; A53T $F_{(1, 20)}$ = 36.42, *P* < 0.0001; interaction $F_{(1, 20)}$ = 0.0062, *P* = 0.9382. Bonferroni's correction for multiple comparisons revealed *P* = 0.9973 for LAG3 WT versus KO in A53T TG and *P* > 0.9808 for LAG3 WT versus KO in A53T WT.

D   Immunofluorescence staining and quantification of Thioflavin S-positive α-synuclein inclusions in HSCs at 5 weeks post-seeding. Scale bars represent 500 μm (overview), and 100 μm (insets). Shown is mean ± SEM, and individual values are shown; *n* = 5–6 cultures per group; two-way-ANOVA revealed for LAG3 $F_{(1, 19)}$ = 1.972, *P* = 0.1763; A53T $F_{(1, 19)}$ = 62.49, *P* < 0.0001; interaction $F_{(1, 19)}$ = 1.586, *P* = 0.2231. Bonferroni's correction for multiple comparisons revealed *P* = 0.1373 for LAG3 WT versus KO in A53T TG and *P* > 0.9999 for LAG3 WT versus KO in A53T WT.

Source data are available online for this figure.

fibrils or FGL1. A possible explanation is that a single α-synuclein fibril binds to multiple LAG3 molecules through electrostatic and/or hydrophobic interactions, resulting in cooperative unspecific interactions that can be captured with ELISA where proteins are immobilized, but not with MDS.

A pulldown experiment confirmed the weak interaction between LAG3 and α-synuclein fibrils. Due to the promiscuous binding of α-synuclein fibrils, their specificity to LAG3 seems questionable. Experiments performed by Mao *et al* (2016) indicate that LAG3$^{-/-}$ reduces α-synuclein fibril binding in cortical neurons by maximally 10% (see figure 1D, in Mao *et al* (2016)). In the original report, mean values in LAG3$^{-/-}$ primary cortical neurons differ only at high α-synuclein fibril-biotin concentrations and display a relatively high standard error (Mao *et al*, 2016), suggesting that there was no biologically important difference. This is not surprising in view of LAG3 not being expressed in neuronal cultures, so its ablation should not have an effect. Our experiments in human neural cultures point in the same direction as we could not identify any significant difference in pS129-positive aggregates upon expression of LAG3, strongly suggesting that other neuronal factors, some of which we identified through a pulldown approach (Shrivastava *et al*, 2015), mediate the uptake and transmission of α-synuclein fibrils.

The negative findings discussed above do not preclude a modulatory role for LAG3 in α-synucleinopathies through other mechanisms. We investigated whether LAG3 ablation changes the propensity of α-synuclein aggregation *in vivo* and, importantly, whether it affects the overall survival of α-synuclein-overexpressing mice. However, our study conducted in A53T human α-synuclein TG mice KO, hemizygous or WT for LAG3 showed that neither lifespan nor deposition of α-synuclein aggregates is significantly changed upon the presence or absence of LAG3, leaving little hope that LAG3 would assume a role in α-synuclein-related pathologies. Experiments with hippocampal organotypic slice cultures inoculated with α-synuclein PFFs did not change this perception to the better and do not seem to converge with the notion that LAG3 contributes to α-synucleinopathy in different A53T human α-synuclein mice (Gu *et al*, 2021).

Parkinson's disease and other pathologies underlying the spread of α-synuclein aggregates are severe neurodegenerative diseases with momentous implications on the lives of patients and their families. More work is needed to understand how α-synuclein is transmitted from cell to cell, to identify selective receptors for propagating forms of α-synuclein and to subsequently enable a targeted treatment. Until then, potential targets need to be rigorously vetted, bearing in mind that attrition of futile approaches is crucial to avoiding high opportunity costs.

# Materials and Methods

## Usage of human material

All experiments and analyses involving samples from human donors were conducted with the approval of the local ethics committee (BASEC-2020-00234), in accordance with the provisions of the Declaration of Helsinki, the Department of Health and Human Services Belmont Report and the Good Clinical Practice guidelines of the International Conference on Harmonisation. Informed consent was obtained from all individuals if applicable. The local ethics committee issued a clearance certificate/declaration of no-objection when fully anonymized human samples were used.

## Animal work

Mice housing were in accordance with the Swiss Animal Welfare Law and in compliance with the regulations of the Cantonal Veterinary Office, Zurich (permits 033/2018, 236/2019). Mice were bred in high hygienic grade facilities and housed in groups of 3–5, under a 12-h light/12-h dark cycle (from 7 am to 7 pm) at 21 ± 1°C, with sterilized food (Kliba No. 3431, Provimi Kliba, Kaiseraugst, Switzerland) and water *ad libitum*. For primary culture, we used 8- to 9-week-old C57BL/6J mice. When needed, two pregnant females (E14) were delivered and housed in the animal facility of the University of Zurich.

A53T-α-synuclein (Thy1-hA53T-αS) (Van Der Putten *et al*, 2000) and LAG3 KO (B6.129S2-*LAG3$^{tm1Doi}$*/J) (Miyazaki *et al*, 1996) animals were bred and kept under specific pathogen-free conditions at the Hertie Institute for Clinical Brain Research in Tübingen, Germany, with sterilized food and water *ad libitum*. For estimation of survival rates, genders were balanced within the groups, whereas only female mice were used for immunological and Thioflavin S staining. LAG3 KO mice were initially purchased from Jackson

Laboratories (Bar Harbor, ME, USA) and crossed to our internal strains to generate the various genotypes used in this study. The experimental procedures were carried out in accordance with the veterinary office regulations of Baden-Wuerttemberg (Germany) and approved by the local Animal Care and Use Committees.

### Antibodies used

See Table 1.

### Antigens used

See Table 2.

### Mouse primary cultures

Primary neuronal cell cultures were prepared from brains of E16-17 mouse embryos. Briefly, hippocampus and cortex were isolated in PBS-Glucose (D-Glucose, 0.65 mg/ml). The tissue was treated with trypsin (0.5% w/v) in PBS-Glucose and dissociated in Neurobasal Medium (NB) supplemented with glutamine (2 mM), 2% B27, 2% N2, 100 U penicillin–streptomycin (P/S) and D-Glucose (0.65 mg/ml). In order to obtain a mixed culture, the medium was supplemented with 2.5% Horse Serum for the first 24 h *in vitro*, while for neuronal-enriched culture, cells were treated with 5 μM cytosine arabinoside (AraC) from DIV 1 to DIV 6. For biochemical experiments, cells were then plated onto poly-D-lysine coated 6-well plate (TPP-92006) at $8 \times 10^5$ cells/cm$^2$. For imaging, cells were plated onto poly-D-lysine-coated chambered coverglass (NuncTM Lab-TekTM - 155411) at $2 \times 10^5$ cells/cm$^2$. For both types of culture, only one-half of the medium was exchanged every 3–4 days. When lentiviral transduction was used to generate LAG3-expressing cultures, cells were transduced at DIV 6 and fixed at DIV 13. Primary glia cell cultures were prepared from postnatal (P4-P6) pups. Briefly, hippocampus and cortex were isolated. The tissue was treated with trypsin (0.5% w/v) in HBSS-Glucose (D-Glucose, 0.65 mg/ml) and triturated with glass pipettes to dissociate tissue in DMEM/F12 (31330038) supplemented with glutamine (2 mM), 5% horse serum and 100 U penicillin–streptomycin, 100 μM non-essential amino acid and 2 mM sodium pyruvate. For biochemical experiments, cells were plated 6-well plate (TPP-92006) at $8 \times 10^5$ cells/cm$^2$. For imaging, cells were plated onto chambered coverglass (NuncTM Lab-TekTM-155411) at $2 \times 10^5$ cells/cm$^2$.

### Synaptoneurosome preparations and region-specific dissections of murine brains

Synaptosome fractions were prepared following the protocol performed as described earlier (De Rossi *et al*, 2020). Briefly, cortices, striata and hippocampi were dissected from the brains of adult C57BL/6J mice and resuspended in cold buffer containing 0.32 M sucrose and 10 mM HEPES at pH 7.4 and centrifuged twice (at 770 ×g) to remove nuclei and large debris, followed by centrifugation at $12,000 \times g$ to obtain the synaptosome fraction. The synaptosomes were washed and pelleted in EDTA buffer to chelate calcium (4 mM HEPES, 1 mM EDTA, pH 7.4, 20 min at $12,000 \times g$). The synaptosomes were resuspended in the lysis buffer for 1 h on ice (20 mM HEPES, 0.15 mM NaCl, 1% Triton X-100, 1% deoxycholic acid, 1% SDS, pH 7.5) and stored at −80°C. Region-specific dissections (cortex, hippocampus, olfactory bulb, corpus callosum, midbrain, striatum, hindbrain, cerebellum) were performed in the same mice and solubilized in 20 mM HEPES, 0.15 mM NaCl, 1% Triton X-100, 1% deoxycholic acid, 1% SDS, pH 7.5 and stored at −80°C.

### Isolation and activation of mouse T lymphocytes

T lymphocytes were isolated from the spleen of C57BL/6J mice using a negative selection isolation kit (EasySep™ Mouse T Cell Isolation Kit, StemCell Technologies). Cells were plated in a 48-well plate (Corning) previously coated with anti-mouse CD3 (clone #17A2, PeproTech) and anti-mouse CD28 (clone #37.51, PeproTech) antibodies and incubated in IMEM medium supplemented with IL-2 at 37°C. After 3 days, the activated T lymphocytes were collected and processed for either RNA extraction or Western blotting. The expression of mLAG3 was checked by Western blotting using mLAG3-transfected HEK cells as positive control.

### Isolation and activation of human T lymphocytes

Fully anonymized residual full blood samples were collected at the Institute of Clinical Chemistry (USZ) using a workflow outlined previously (preprint: Emmenegger *et al*, 2020), and peripheral blood mononuclear cells (PBMCs) were isolated using a Ficoll gradient. PBMCs were resuspended in RPMI medium supplemented with 10% FBS, 1× P/S, 1× non-essential amino acids (Gibco) and 1× sodium pyruvate (Gibco), to reach a concentration of 2 million cells per ml. 2 ml cell suspension/well was added into a 24-well plate. 1 ml medium was removed every second day, and 1 ml RPMI medium with supplements as listed above as well as with 100 international units IL-2 and 2 μg/ml phytohemagglutinin (PHA) was added. T cells were cultured between 7 and 9 days. Finally, the medium was removed, 100 μl lysis buffer (50 mM Tris, pH 8, 150 mM NaCl, 1% Triton, protease and phosphatase inhibitors) was added to each well and multiple wells were pooled. Following a centrifugation step (16,000 *g*, 20 min, 4°C), the supernatant was transferred to a clean tube and the protein concentration measured using the BCA assay.

### Lentivirus preparation, transduction and SH-SY5Y cell line generation

Using NEBuilder HiFi DNA Assembly Cloning Kit (NEB #E5520), human or mouse LAG3 cDNA sequence (GeneCopoeia #EX-Z5714-M02 and #EX-Mm03576-M02) was inserted into an autoregulatory, all-in-one TetON cassette (AutoTetON; Hruska-Plochan *et al*, 2021), which was previously inserted into a pLVX lentiviral transfer vector (Clontech # 632164), while deleting CMV-PGK-Puro, generating Auto-hLAG3 and Auto-mLAG3 lentiviral transfer vectors. These were then packaged into lentivirus via co-transfection with CMV-Gag-Pol (Harvard #dR8.91) and pVSV-G (Clontech, part of #631530) plasmids into production HEK293T cells adapted to grow in serum-free conditions (OHN media Hruska-Plochan *et al*, 2021), which reduces the expression of the GOI from the transfer vector as well as it eliminates serum carry over into the supernatant. Medium was changed the following morning, and supernatants were then collected 48 h post-transfection (36 h postmedia change),

**Table 1.   Details on antibodies used in current study.**

| Name | Immunogen | Experiment |
|---|---|---|
| 4-10-C9 mouse monoclonal antibody (MABF954, Merck Millipore) | Mouse LAG3 | Mouse Western Blot, Mouse LAG3 sandwich ELISA, indirect ELISA |
| LSB15026 rabbit polyclonal antibody (LSBio) | Mouse LAG3 | Mouse Western Blot, indirect ELISA |
| rat mAb C9B7W | Mouse LAG3 | Mouse LAG3 sandwich ELISA, indirect ELISA |
| LAG3 (D2G4O™) XP® Rabbit mAb #15372 | Human LAG3 | Human Western Blot, immunofluorescence, human LAG3 sandwich ELISA, indirect ELISA |
| Anti-LAG3 antibody [17B4] (ab40466) | Human LAG3 | Human Western Blot, immunofluorescence, indirect ELISA |
| Relatlimab anti-LAG3 antibody (BMS-986016) | Human LAG3 | Human LAG3 sandwich ELISA, indirect ELISA |
| LAG525 (Novartis) | Human LAG3 | Indirect ELISA |
| Mouse mAb human LAG3 (Clone 874501) | Human LAG3 | Indirect ELISA |
| Anti-Alpha-synuclein (phospho S129) antibody [P-syn/81A] (ab184674) | pS129 α-synuclein | Immunofluorescence |
| Recombinant Anti-Alpha-synuclein (phospho S129) antibody [EP1536Y] (ab51253) | pS129 α-synuclein | Immunofluorescence |
| Recombinant Anti-Alpha-synuclein antibody [MJFR1] (ab138501) | α-synuclein | ELISA with fibrils, immunofluorescence, Microfluidic diffusional sizing |
| Polyclonal rabbit anti-ASC antibody AL177 (AdipoGen, #AG-25B-0006) | ASC | ELISA with fibrils |
| Anti-Glial Fibrillary Acidic Protein (GFAP) (ab53554) | GFAP | Immunofluorescence analysis of mouse primary cultures |
| Anti-MAP2 antibody (ab5392) | MAP2 | Immunofluorescence analysis of mouse primary cultures |
| Anti-Iba1 (WA3 019-19741) | Iba1 | Immunofluorescence analysis of mouse primary cultures |
| HRP Donkey anti-rat IgG (H + L), 712-035-153, Jackson | | Secondary antibody, ELISA, Western blot |
| HRP Goat anti-Rabbit IgG (H + L), 111-035-045, Jackson | | Secondary antibody, ELISA, Western blot |
| HRP Goat anti-mouse IgG (H + L), 115-035-003, Jackson | | Secondary antibody, ELISA, Western blot |
| Peroxidase AffiniPure Goat Anti-Human IgG, Fcγ Fragment Specific, Jackson, 109-035-098 | | Secondary antibody, ELISA, Western blot |
| Donkey anti-rabbit AF568 (Invitrogen #A10042) | | Secondary antibody, immunofluorescence |

**Table 1**  (continued)

| Name | Immunogen | Experiment |
|---|---|---|
| Donkey anti-mouse AF488 (Invitrogen #A-21202) | | Secondary antibody, immunofluorescence |
| Donkey anti-chicken AF647 (Jackson ImmunoResearch #JAC703-606-155) | | Secondary antibody, immunofluorescence |

**Table 2.   Details on antigens used in current study.**

| Name | Experiment |
|---|---|
| Human APOE3, PeproTech (#350-02) | ELISA with fibrils |
| Human ASC-C-his, produced by Matthias Geyer (Bonn) | ELISA with fibrils |
| Human LAG3$_{23-450}$, ACROBiosystems (#LA3-H5222) | ELISA with fibrils, antibody testing |
| Human LAG3$_{23-434}$, BonOpusBio (#CJ91) | MDS |
| Mouse LAG3$_{24-442}$, ACROBiosystems (#LA3-M52H5) | ELISA with fibrils, antibody testing |
| Natural Ara h 2 (NA-AH2-1), Light roasted peanut flour (Runner cultivar), Indoor Biotechnologies | ELISA with fibrils |
| Human recombinant PrP$^C$, produced in-house (Adriano Aguzzi) | ELISA with fibrils |
| Bovine serum albumin (BSA), Thermo Scientific | ELISA with fibrils, MDS |
| FGL1 Protein, Human, Recombinant, 13484-H08B, Sino Biological | MDS |
| Human α-synuclein, produced in-house (Kelvin Luk) | ELISA with fibrils |
| Human CD4, ACROBiosystems (#LE3-H5228) | ELISA with fibrils, MDS |
| MTBD tau, produced in-house (Adriano Aguzzi) | ELISA with fibrils |

centrifuged at (500 $g$, 10 min, 4°C), filtered through Whatman 0.45-μm CA filter (GE #10462100) and concentrated using Lenti-X™ Concentrator (Takara #631232) according to producer datasheet (overnight incubation). The resulting lentiviral pellets were then resuspended in complete neuronal maturation media or OHN media to achieve 10× concentrated LV preparations, which were titrated using Lenti-X™ GoStix™ Plus (Takara #631280). Auto-hLAG3 had a GoStix Value (GV) of 6496 and Auto-mLAG3 had GV of 12839.

### Transduction of NSC-derived human neural cultures

Differentiated human neural cultures (2 months old) were transduced with 250 μl of Auto-hLAG3 LV and 3 μg/ml of polybrene (Sigma-Aldrich #TR-1003-G) per well of a 6-well plate. Medium was exchanged completely the following day. hLAG3 expression was induced by 1 μg/ml of Doxycycline (DOX; Clontech #631311).

### Transduction of SH-SY5Y cells to generate SH-10 line inducibly expressing hLAG3

SH-SY5Y human neuroblastoma cells (Sigma-Aldrich #94030304) at P11 were transduced in a 6-well plate using 300 μl of Auto-hLAG3

LV pre-incubated with 30 µl of Lenti-X™ Accelerator (Clontech #631256) for 30 min. The LV mixture was added onto the cells, evenly spread over the whole well and put into incubator onto a neodymium magnetic sheet (supermagnete #NMS-A4-STIC; adhesive force of 450 g/cm²) for 10 min. Cells were then removed from the incubator, LV mixture was completely removed, and fresh SH media added. Cells were kept until confluency and then sub-cultured with the neodymium magnet sheet kept under the plate. To remove remaining magnetic beads, cell pellet was resuspended in 1.5 ml of SH media, pipetted into 1.5-ml Eppendorf tube and inserted into DynaMag™-2 Magnet (Invitrogen #12321D). Cell suspension was then removed from the tube leaving all magnetic beads in the tube. Quantification of P11+1 SH-10 cells showed that 76% of cells were expressing hLAG3 upon induction by 1 µg/ml of Doxycycline (DOX; Clontech #631311). This decreased to 59% in P11+2 and remained around 50% in the following passages. P11+3 was used for all experiments except for the CO-IP, where a later passage was used.

## Maintenance of NSC-derived neural cultures

Human neural stem cells derived from iPSCs using manual selection based on colony morphology (Bohaciakova *et al*, 2019)—iCoMoNSCs (Hruska-Plochan *et al*, 2021)—were differentiated for 2 months resulting in functional neural networks (Hruska-Plochan *et al*, 2021). Shortly, iCoMoNSCs were plated onto Matrigel-coated 6-well plates and grown in NSC media until reaching confluency. Media was then changed to D3 differentiation media, which was replaced for maturation media at 4 weeks of differentiation. For imaging experiments, cultures were dissociated into single-cell suspension using Papain Dissociation System (Worthington #LK003150), passed through 70-µm cell strainer (Falcon #07-201-431), resuspended in maturation media and re-plated into 96-well imaging plate (Greiner Bio-One #655090) at 120,000 cells per well as counted by CASY Cell Counter (Innovatis AG).

## scRNAseq in NSC-derived neural cultures—Sample preparation

Differentiated human neural cultures (3 months old) were dissociated into single-cell suspension using Papain Dissociation System (Worthington #LK003150), passed through 70-µm and 40-µm cell strainers (Falcon #07-201-431 and #07-201-430) and resuspended in HIBE++ media (Hibernate™-E Medium (Gibco #A1247601) supplemented with EDTA (1mM final; Invitrogen #AM9260G), HEPES (10 mM final; Gibco #15630080), with 1× B27+ supplement (Gibco #17504-044), 1X N2 supplement (Gibco #17502-048); 1× GlutaMAX (Gibco #35050-061), BDNF (PeproTech #450-02), GDNF (Alomone labs #G-240), CNTF (Alomone labs #C-240), NT-3 (PeproTech #450-03) and IGF-1 (Stem Cell #78022) all at 20 ng/ml to 1,000 cells per µl using to CASY Cell Counter (Innovatis AG).

## scRNAseq in NSC-derived neural cultures using 10X Genomics platform

The quality and concentration of the single-cell preparations were evaluated using an haemocytometer in a Leica DM IL LED microscope and adjusted to 1,000 cells/µl. 10,000 cells per sample were loaded into the 10× Chromium controller, and library preparation

was performed according to the manufacturer's indications (Chromium Next GEM Single Cell 3′ Reagent Kits v3.1 protocol). The resulting libraries were sequenced in an Illumina NovaSeq sequencer according to 10× Genomics recommendations (paired-end reads, R1 = 28, i7 = 8, R2 = 91) to a depth of around 50,000 reads per cell. The sequencing was performed at Functional Genomics Center Zurich (FGCZ).

## scRNAseq data analysis in NSC-derived neural cultures—Cell clustering and differential expression on each sample

The fastq files were aligned to the Homo Sapiens reference sequence (build GRCh38.p13) taken from Ensembl. After the alignment, each observed barcode, UMI, gene combination was recorded as a UMI count matrix that was then filtered to remove low RNA content cells or empty droplets using the CellRanger software (v3.0.1). Starting from this matrix we used the R package Seurat (version 3.1.5) (Butler *et al*, 2018) to perform the following downstream analyses per sample: genes and cells filtering, normalization, feature selection, scaling, dimensionality reduction, clustering and differential expression. We started by filtering out genes that did not obtain at least 1 UMI count in at least 3 cells, and discarded cells for which fewer than 1,500 genes or more than 8,000 genes were detected and also those that had a mitochondrial genome transcript ratio > 0.12. After this, the data were normalized using a global-scaling normalization method that normalizes the feature expression measurements for each cell by the total expression, multiplies this by a scale factor (10,000 by default) and log-transforms the result. We next calculated a subset of 2,000 features that exhibited high cell-to-cell variation in the data set. Using as input these variable features, we performed PCA on the scaled data. Since Seurat clusters cells based on their PCA scores, we used a heuristic method called "Elbow plot" to determine how many principal components (PCs) we needed to capture the majority of the signal. In this way, the cells were clustered with an unsupervised graph-based clustering approach using the first 30 first PCs and a resolution value of 0.5. Clusters were visualized using t-distributed Stochastic Neighbor Embedding of the principal components (spectral t-SNE) or Uniform Manifold Approximation and Projection for Dimension Reduction (UMAP) as implemented in Seurat. We found positive markers that defined clusters compared to all other cells via differential expression. The test we used was the Wilcoxon rank-sum test which assesses separation between the expression distributions of different clusters. Genes with an average, at least 0.25-fold difference (log-scale) between the cells in the tested cluster and the rest of the cells, and an adjusted $P < 0.01$ were declared as significant. Cell cycle phases were predicted using a function included in the scran R package (Lun *et al*, 2016) that scores each cell based on expression of canonical marker genes for S and G2/M phases. To visually, qualitatively and quantitatively interrogate expression of particular genes in all cells amongst all clusters via t-SNE, UMAP, violin and distribution plots, we used a custom-built Shiny application.

## Propagation experiment in NSC-derived neural cultures: PFF-induced synuclein aggregation and aggregates quantification

Human neural cultures were transduced as described above. 4 days later, neural cultures were sub-cultured into 96-well plates as

described above. hLAG3 expression was then induced by addition of 1 µg/ml of Doxycycline (DOX; Clontech #631311) 3 weeks (4 weeks of DOX on conditions) or 5 weeks (2 weeks of DOX on conditions) later. 4 days post-hLGA3 expression activation, 500 nM of human exogenous α-synuclein pre-formed fibrils (PFFs) produced from monomeric recombinant α-synuclein by Kelvin Luk Lab as per (Volpicelli-Daley et al, 2014) was added to half of the wells (resulting in 8 conditions: 2 weeks of DOX$^{+/-}$/ PFFs$^{+/-}$ ($n = 4$ wells per condition) and 4 weeks of DOX$^{+/-}$/ PFFs$^{+/-}$ ($n = 5$ wells per condition). 50 µl of fresh media was added 4 days and then again 6 days later, and 50% of media was exchanged on day 8 of the treatment and then three times a week until the end of the experiment while refreshing the DOX at 1 µg/ml.

### Propagation experiment in NSC-derived neural cultures: Fixation, immunofluorescence and imaging

Neural cultures were fixed with pre-warmed 16% methanol-free formaldehyde (Pierce #28908) pipetting it directly into the culture media, diluting it to 4% final, and incubated for 15 min at room temperature. Cells were then washed once with PBS (Gibco #10010015) for 10 min, once with PBS with 0.2% Triton X-100 (TX; Sigma #T9284) washing buffer (WB) for 10 min and then blocked with 10% normal donkey serum (Sigma-Aldrich #S30-M), 0.2% TX in PBS blocking buffer filtered via Stericup (Millipore #S2GPU02RE) for 30 min at room temperature (RT). Primary antibodies were then diluted in blocking buffer (81A 1:500; D2G40 1:500; MAP2 Abcam #ab5392; EP1536Y 1:500; 17B4 1:500; MJFR1 1:1,000) and left incubated overnight (ON) at 4°C on an orbital shaker. Cells were then washed 3 × 15 min in WB at RT, and secondaries were then diluted in blocking buffer (Donkey anti-rabbit AF568 (Invitrogen #A10042) 1:750; Donkey anti-mouse AF488 (Invitrogen #A-21202) 1:750; Donkey anti-chicken AF647 (Jackson Immuno Research #JAC703-606-155) 1:500) and incubated for 1.5 h at RT. Cells were then again washed 3 × 15 min in WB at RT with DAPI (Thermo Scientific #62248) diluted to 1 ug/ml in the final WT wash. Cells were finally washed 1 × 15 min in PBS at RT, and PBS was then added to the wells to store the stained cells at 4°C. Human neural cultures were imaged using GE InCell Analyzer 2500 HS widefield microscope for quantification (40× air objective; 2D acquisition; 182 fields of view per well; 50 µm separation to avoid counting cells twice) or with Leica SP8 Falcon inverted confocal for high power, high-resolution microscopy (63× oil objective; 2,096 × 2,096 pixels at 0.059 µm/pixel, approx. 20–30 z-steps per stack at 0.3 µm). Laser and detector setting were kept same for each staining combination and all imaged conditions. Huygens professional (Scientific Volume Imaging, Hilversum, Netherlands) was then used to deconvolute the stacks, and the deconvoluted images were further post-processed in fiji to produce a flattened 2D pictures (Z-projection) for data visualization.

### Propagation experiment in NSC-derived neural cultures: signal quantification

Trained ilastik (Berg et al, 2019) algorithms were used to segment the pixels (positive versus background) of 81A, EP1536Y and MAP2 stainings. Segmented pictures (182 images per well, per channel) were then exported and the total signal quantified in fiji via batch processing using a custom macro. Sum of all positive pixels of all 182 fields of view per well for 81A and EP1536Y was then expressed as % of total MAP2 area occupied by 81A or EP1536Y signal. Statistical analysis was performed in Prism (GraphPad San Diego, CA, USA), and one-way ANOVA followed by Tukey's multiple comparison test was applied on the datasets.

### Propagation experiment in NSC-derived neural cultures: PFF preparation and treatment

PFFs were sonicated before their addition to the neural cultures as follows: stock PFFs at 5 mg/ml were diluted in PBS (Gibco # 10010015) inside of a standard 0.5-ml Eppendorf tube to 0.1 mg/ml reaching 250 µl final volume. The diluted PFFs were then sonicated using ultrasonic processor (Qsonica #Q500A-110) equipped with a cup horn (Qsonica #431C2) allowing for indirect sonication via high-intensity ultrasonic water bath using the following settings: Amplitude 40%, 30 cycles of 2-s on, 2-s off (resulting in total run time of 120 s; 60-s sonication, 60-s off). Ice-cold water was freshly poured into the cup immediately before the sonication. 210 µl of sonicated PFFs was further diluted in 2790 µl of pre-warmed complete maturation media (reaching 500 nM) and added to the neural cultures (250 µl per well; all spent media was removed before treatment).

### snRNA sequencing in human frontal cortex

Single nuclei RNA sequencing (snRNAseq) analysis of human frontal cortex was completed as previously described (Saez-Atienzar et al, 2021). In brief, frozen frontal cortex tissue samples from 16 donors in the North American Brain Expression Consortium study series (dbGaP Study Accession: phs001300.v1.p1, (Myers et al, 2007)) were obtained from the University of Maryland Brain and Tissue Bank through the NIH NeuroBioBank. Nuclei were isolated from 21 total samples from the 16 donors by ultracentrifugation through a sucrose gradient, and the Chromium Single Cell Gene Expression Solution v3 (10× Genomics) was used to construct snRNAseq libraries. Libraries were pooled and sequenced using the Illumina NextSeq 550 System. The resulting FASTQ files were aligned and counted using Cell Ranger v3 (10× Genomics). The 21 data sets were integrated, and clustering analysis was performed using Seurat v3.1 (Stuart et al, 2019) in R. Cell clusters comprised of a total of 161,225 nuclei were manually assigned cell type identities based on differential expression of known cell type marker genes. The cell type abbreviations are as follows: "EC" = endothelial cell; "AST" = astrocyte; "MGL" = microglia; "OPC" = oligodendrocyte precursor cell; "ODC" = oligodendrocyte; "InN" = inhibitory neuron; "ExN" = excitatory neuron.

### scRNA sequencing in mouse mesencephalon and striatum

Single-cell suspensions were generated from the midbrain and striatum of 1-year-old animals. All the animals have received a single intra-striatal injection with either lipopolysaccharide (LPS) or phosphate-buffered saline (PBS) and were sacrificed 48 h after surgery as described previously (Russo et al, 2019). Brain areas were dissociated using Adult Brain Dissociation Kit (Miltenyi Biotec #130-107-677) following by myelin removal with 20-µl myelin

removal beads (Miltenyi Biotec #130-096-733). Cell concentration and viability of the single-cell solutions were determined on LunaFL cell counter (Logos Biosystems) and adjusted to 1,000 cells per μl. Preparations had more than 85% viability and were kept on ice prior to single-cell encapsulation and library preparation. For encapsulation, 8,500 cells per sample were loaded into a Single-Cell 3′ Chip of the 10× Chromium controller and libraries were generated using 10xGenomics (Chromium Next GEM Single Cell 3′ Reagent Kits v3.1 protocol). The resulting libraries were examined by Agilent Bioanalyzer 2100 using a High Sensitivity DNA chip (Agilent). Libraries were sequenced at NIH Intramural Sequencing Center (NISC, Bethesda, USA) on an Illumina NovaSeq600 sequencer, using S4 flow cell format according to 10× Genomics recommendations (2 × 150 bp, paired-end reads, Read1 = 28, i7 Index = 8, Read2 = 91) with an estimated read depth of 100,000 reads per cell.

Sequencing data were analysed using the Cell Ranger Pipeline (version v3.1.0) to perform quality control, sample demultiplexing, barcode processing, alignment and single-cell 3′ gene counting. Genome Reference Consortium Mouse Build 38 (GRCm38-mm10) Mus musculus reference transcriptome was used for alignment. Single-cell data from PBS- or LPS-treated animals were combined into a single Seurat object and filtered for genes detected in more than three cells and for cells that had more than 500 genes (features). These data sets were normalized using SCTransform v0.2.1 and integrated by pair-wise comparison of anchor gene expression within the Seurat package v3.1 in R (Butler *et al*, 2018). Shared nearest neighbour-based clustering was used to identify distinct cell clusters, which were then manually assigned cell type identities based on differential expression of known cell type marker genes.

### iPSC-derived dopaminergic cultures

iPSCs were differentiated as previously described (Bengoa-Vergniory *et al*, 2020). Dopaminergic neuronal cultures from control lines and GBA N370S PD patients were harvested at 35DIV and stained for TH (ab152, Millipore) or harvested for Western blotting.

### Western blotting

Whole mouse brains were homogenized in RIPA buffer, whereas mouse primary cortical cultures and activated T lymphocytes were lysed in lysis buffer (50 mM Tris, 150 mM NaCl, 1% Triton X-100) containing protease and phosphatase inhibitors (Roche, New York, NY, USA). Total protein contents were quantified using the BCA assay (Pierce). For Western blotting, total proteins were separated on SDS–polyacrylamide gels (4–12%, Invitrogen) and transferred onto PVDF membranes. Membranes were blocked in 5% SureBlock (LubioScience) for 1 h at RT and incubated overnight at 4°C with anti-LAG3 mouse monoclonal antibody 4-10-C9 (1:1,000). Membranes were washed three times with PBS-Tween and incubated with goat-anti-mouse IgG conjugated with horseradish peroxidase (1:10,000). The acquisition was performed using ECL Crescendo substrate (Merck Millipore) and imaged with Fusion Solo S (Vilber). After the acquisition, the membranes were incubated for 30 min at RT with anti-β-actin (1:10,000, A3854 Sigma-Aldrich), which was used as a loading control.

For IP enrichment, 500 μg of total proteins of mouse brain homogenates and primary cultures and 10 μg of total proteins of activated

T lymphocytes were incubated overnight with 50 μl of Dynabeads™ (Thermo Fisher) previously conjugated with 4-10-C9 antibody. Beads were washed three times with PBS, resuspended in Laemmli buffer and dissociated by boiling for 5 min at 95°C. The supernatants were loaded onto SDS–polyacrylamide gels and processed as described above. Target antigens were detected using an anti-LAG3 rabbit polyclonal antibody (1:1,000, B15026) and goat-anti-rabbit IgG conjugated to horseradish peroxidase (HRP) (1:1,000).

### Mouse LAG3 Sandwich ELISA

High-binding 384-well SpectraPlates (Perkin Elmer) were coated with mouse mAb 4-10-C9 at a concentration of 1 μg/ml in sterile PBS and at a volume of 20 μl/well. The plates were incubated for 1 hr at 37°C and washed 3× with 1× PBS 0.1% Tween-20. Plates were then blocked with 5% milk in 1× PBS 0.1% Tween-20 (40 μl/well) and incubated for 1 h at RT. The blocking buffer was subsequently removed. The cell lysates/organ homogenates were added at a total protein concentration of 1,000 μg/ml (or, if indicated, at 500 μg/ml due to insufficient sample concentration), 20 μl/well, in quadruplicates and mouse recombinant LAG3$_{24–442}$, was added at 5 μg/ml at highest concentration, 40 μl/well into the first well of the dilution series, and a 15-fold 1:2 serial dilution of the recombinant proteins was performed (end volume for each well: 20 μl). The buffer used for sample dilution: 1% milk in 1× PBS 0.1% Tween-20. The plates were incubated for 1 h at RT, followed by a 3× wash with 1x PBS 0.1% Tween-20. To detect the presence of LAG3, we used rat mAb C9B7W (for mouse LAG3), at 1 μg/ml and at 20 μl/well. The plates were incubated for 1 h at RT, followed by a 5× wash with 1× PBS 0.1% Tween-20. To detect the presence of the detection antibody, we used HRP-anti-rat IgG (1:2,000, for mouse LAG3, HRP Donkey anti-rat IgG (H + L), 712-035-153, Jackson), at volume of 20 μl/well. The plates were then incubated for 1 h at RT and washed 3× with 1× PBS 0.1% Tween-20. TMB was dispensed at a volume of 20 μl/well, followed by a 5-min incubation, and the stop solution (0.5 M H$_2$SO$_4$) was added at a volume of 20 μl/well. The absorbance at 450 nm was read on the EnVision (Perkin Elmer) reader. Finally, the measured optical densities were interpolated (independently for all replicates) using the mouse recombinant LAG3 standards (values that are in the linear range) and LAG3 expression per μg total protein was depicted for all samples.

### RNA extraction and RT–qPCR

Mouse primary cultures were collected and lysed in lysis buffer and TRIzol LS Reagent in a ratio 1:3. Samples were supplemented with 0.2 ml of chloroform per 1 ml of TRIzol and centrifugated. The aqueous phase was collected, and the RNA was precipitated with subsequent addition of isopropanol and ice-cold 75% ethanol. Pellets were resuspended in ultrapure, RNAse-free water and quantified at the NanoDrop. Genomic DNA elimination and reverse transcription were performed using QuantiTect Reverse Transcription Kit (Qiagen) according to manufacturer's instructions.

To generate a standard curve for transcript quantification, a commercial plasmid encoding full-length mouse LAG3 CDS (GeneCopoeia, Mm0357-02) was used as standard. The number of DNA molecules in 1 μl was calculated using the following formula:

$$\frac{\text{copy number}}{\mu l} = \frac{\text{ng}}{\mu l} * \frac{6.022 * 10^{23}}{(\text{number bp vector} + \text{LAG3 amplicon bp}) * 10^9 * 650}$$

Serial dilutions of the standard containing known numbers of molecules and 25 ng of total cDNA of the samples were amplified in RT–qPCR using mouse LAG3-specific primers (Liu et al, 2018). The number of transcripts per ng of RNA was calculated by interpolating the threshold cycle values of the samples into the standard curve and then dividing by the ng of RNA used in the RT–qPCR.

### ELISA with fibrils

High-binding 384-well SpectraPlates (Perkin Elmer) were coated with proteins of interest at a concentration of 1 µg/ml in sterile PBS and at a volume of 20 µl/well. The plates were incubated for 1 h at 37°C and washed 3× with 1× PBS 0.1% Tween-20. Plates were then blocked with 5% SureBlock (Lubio) in 1× PBS 0.1% Tween-20 (40 µl/well) and incubated for 1 h at RT. The blocking buffer was subsequently removed. Monomer equivalents of fibrils (α-synuclein or ASC) were added at a concentration of 0.125 µM, 20 µl/well and a 13-fold 1:2 serial dilution was performed (end volume for each well: 20 µl). The buffer used for sample dilution: 1% SureBlock in 1× PBS 0.1% Tween-20. The plates were incubated overnight at 4°C, followed by a 5× wash with 1× PBS 0.1% Tween-20. In a control experiment, monomer equivalents of fibrils (α-synuclein or ASC) were coated onto high-binding 384-well SpectraPlates (Perkin Elmer) as serial dilutions (13-fold 1:2 serial dilution), starting at a concentration of 0.125 µM, in sterile PBS and at a volume of 20 µl/well, incubated for 1 h at 37°C, washed 3× with 1× PBS 0.1% Tween-20 and blocked with SureBlock. Next, antibodies directed against the fibrils or monomeric proteins were added at 1 µg/ml and at 20 µl/well, in sample buffer. The plates were incubated for 1 h at RT, followed by a 3× wash with 1× PBS 0.1% Tween-20. To detect the presence of the detection antibody, we used HRP-anti-rabbit IgG (1:2,000, HRP Goat anti-Rabbit IgG (H + L), 111-035-045, Jackson), at volume of 20 µl/well. The plates were then incubated for 1 h at RT and washed 3× with 1× PBS 0.1% Tween-20. TMB was dispensed at a volume of 20 µl/well, followed by a 5-min incubation, and the stop solution (0.5 M $H_2SO_4$) was added at a volume of 20 µl/well. The absorbance at 450 nm was read on the EnVision (Perkin Elmer) reader, and the respective curves were plotted in GraphPad Prism.

### Microfluidic diffusional sizing

Microfluidic diffusional sizing (MDS) measurements were performed as reported previously (Arosio et al, 2016; Wright et al, 2018; Scheidt et al, 2019). Briefly, microfluidic devices were fabricated in polydimethylsiloxane (PDMS) applying standard soft-lithography methods and subsequently bonded onto a glass microscopy slide after activation with oxygen plasma. Samples and buffer were loaded onto the chip from sample reservoirs connected to the inlets and the flow rate controlled by applying a negative pressure at the outlet using a glass syringe (Hamilton, Bonaduz, Switzerland) and a syringe pump (neMESYS, Cetoni GmbH, Korbussen, Germany). Imaging was performed using a custom-built inverted epifluorescence microscope supplied with a charge-coupled-device camera (Prime 95B, Photometrics, Tucson, AZ, USA) and brightfield

LED light sources (Thorlabs, Newton, NJ, USA). Lateral diffusion profiles were recorded at 4 different positions along the microfluidic channels. Varying fractions of unlabelled ligands were added to a solution containing labelled receptors of concentrations varying between 10 nM and 10 µM, and PBS (containing 0.01% Tween-20, SA) was added to give a constant volume of 30 µl. The samples were incubated at room temperature for 60 min, and the size of the formed immunocomplex was determined through measuring the hydrodynamic radius, $R_h$, with microfluidic diffusional sizing, as described above. Dissociation constants ($K_D$) were determined using the Langmuir binding isotherm, as previously reported (Scheidt et al, 2019),

$$R_h = \left( \frac{[L]_{tot} + \alpha[X]_0 + K_D - \sqrt{(\alpha[L]_{tot} + [X]_0 + K_D)^2 - 4\alpha[L]tot[X]_0}}{2} \right) \frac{\Delta R}{\alpha[X]_0} + R_0$$

with $[L]_{tot}$ the concentration of ligand added, $[X]_0$ the concentration of labelled protein, $K_D$ the dissociation constant, $\alpha$ the stoichiometric binding ratio, $\Delta R$ the difference in radius between fully bound and fully unbound and $R_0$ the radius of fully unbound labelled protein.

### Co-immunoprecipitation

SH-SY5Y cells expressing human LAG3 under the control of a doxycycline-inducible promoter were plated in 6-well plates and treated with doxycycline (1 µg/ml) for 72 h to induce the expression of LAG3. Monomeric and fibrillar α-synuclein (1 µM) and PBS (negative control) were added to the culture medium and incubated for 3 h at 37°C. Cells were lysed in lysis buffer (50 mM Tris, pH 8, 150 mM NaCl, 1% Triton, protease and phosphatase inhibitors), and 500 µg of total proteins was further processed. 30 µl was collected from each sample and used as control inputs. Samples were pre-incubated with Dynabeads Protein G to eliminate any possible material that could bind unspecifically to the beads. Subsequently, samples were incubated with Dynabeads functionalized with either anti-α-synuclein or anti-LAG3 antibody and left overnight on a rotor wheel at 4°C. The day after, beads were collected, resuspended in Laemmli buffer and boiled for 5 min to dissociate the precipitated fractions. Supernatants were then loaded onto an SDS–PAGE and processed as described above.

### Immunofluorescence

Primary cultures were fixed at DIV 12-14 with 4% PFA supplemented with 4% sucrose, followed by incubation in 10% donkey serum + 0.2% Triton X-100 for 1 h at room temperature. For immunostaining, antibodies were diluted in blocking buffer (anti-MAP2 1:1,000; anti-GFAP 1:1,000; anti-LAG3 4-10-C9 1:1,000; anti-Iba1 1:1,000) and incubated overnight at 4°C. Samples were then washed 3 times with PBS and incubated with fluorescently conjugated secondary antibodies (1:400) and DAPI (1:10,000).

### Preparation and treatment of hippocampal slice cultures

HSCs were prepared from pups at postnatal days 4–6 (P4-6) according to previously published protocols (Novotny et al, 2016). After

decapitation, the brains of the pups were aseptically removed, and the hippocampi were dissected and cut perpendicular to the longitudinal axis into 350-μm sections with a tissue chopper. Carefully selected intact hippocampal sections were transferred into petri dishes containing ice-cold buffer solution (minimum essential medium (MEM) supplemented with 2 mM GlutaMAX™ at pH 7.3). Three sections were placed onto a humidified porous membrane in a well of a 6-well plate with 1.2 ml culture medium (20% heat-inactivated horse serum in 1× MEM complemented with GlutaMax™ (1 mM), ascorbic acid (0.00125%), insulin (1 μg/ml), $CaCl_2$ (1 mM), $MgSO_4$ (2 mM) and D-glucose (13 mM) adjusted to pH 7.3). HSCs were kept at 37°C in humidified $CO_2$-enriched atmosphere. The culture medium was changed three times per week. HSCs were kept for 10 days without any experimental treatment. At day 10, 1 μl of PFF (5 μg/μl) was pipetted on top of each culture.

### α-Synuclein PFF preparation

We used protocols developed by the Melki and the Luk laboratories. In the Melki protocol, expression in *E. coli*, purification and quality control of human recombinant monomeric wt α-synuclein was done as described (Bousset *et al*, 2013). For fibril formation, soluble wt α-synuclein was incubated in Tris–HCl buffer (50 mM Tris–HCl, pH 7.5, 150 mM KCl) at 37°C under continuous shaking for 5 days and formation of fibrils was assessed with Thioflavin T. The fibrils were quality checked by transmission electron microscopy after negative staining before and after fragmentation. Their limited proteolytic pattern was also assessed (Bousset *et al*, 2013). The average size of the fibrils after fragmentation $47 \pm 5$ nm was derived from length distribution measurements, and their average molecular weight (16,200 Da) was derived from analytical ultracentrifugation sedimentation velocity measurements. The fibrils (350 μM) were aliquoted (6 μl per tube), flash-frozen in liquid nitrogen and stored at –80°C.

In the Luk protocol, expression in *E. coli*, purification and quality control of human recombinant monomeric wt α-synuclein and assembly into PFFs was done as described (Luk *et al*, 2009; Volpicelli-Daley *et al*, 2014). α-synuclein was expressed in *E. coli* and purified to reach a final concentration of monomeric α-synuclein of around 30 mg/ml. Monomeric α-synuclein in 1.5-ml microcentrifuge tubes at a volume of 500 μl and a final concentration of 5 mg/ml was assembled into α-synuclein PFFs by shaking for 7 days at 1,000 RPM in a thermomixer at 37°C. The quality of the fibrils was assessed as detailed earlier (Volpicelli-Daley *et al*, 2014).

### Analysis of human α-synuclein expression in CSF of A53T LAG3$^{-/-}$ and LAG3$^{+/+}$ animals using SIMOA

Samples of the cerebrospinal fluid (CSF) were collected in a standardized manner adapted from the methodology of the Alzheimer's Association external quality control programme used for human CSF (Mattsson *et al*, 2011) as described previously (Schelle *et al*, 2017). Sampling was performed under a dissecting microscope. Mice were deeply anaesthetized with a mixture of ketamine (100 mg/kg) and xylazine (10 mg/kg) and placed on a heat pad to maintain a constant body temperature during the whole procedure.

After incision of the overlying skin and retraction of the posterior neck muscles, the dura mater covering the cisterna magna was carefully cleaned with cotton swabs, PBS and ethanol. In case of microbleeds in the surrounding tissue, hemostyptic gauze (Tabotamp, Ethicon) was used to keep the dura mater spared from any blood contamination. Finally, the dura was punctured with a 30G needle (BD Biosciences) and CSF was collected with a 20 μl gel loader tip (Eppendorf, shortened), transferred into 0.5-ml polypropylene tubes (Eppendorf) and placed on ice. Typically, a total volume of 15–25 μl was collected per mouse. The samples were centrifuged for 10 min at 2,000× *g*, visually inspected for any pellet revealing blood contamination, aliquoted (5 μl) and stored at –80°C until further usage. Samples with suspected blood or cell contamination were not used for any further experiments. CSF α-synuclein concentrations were measured by Single Molecule Array (SIMOA) technology using the SIMOA Human Alpha-Synuclein Discovery Kit according to the manufacturer's instructions (Quanterix, Billerica, MA, USA). Mouse CSF samples were diluted 1:300 with Alpha-Synuclein Sample Diluent before the measurement and analysed on a SIMOA HD-1 Analyzer in duplicates.

### Histology and Immunohistochemistry

Brains were cut in 25 μm-thick sagittal sections on a freeze-sliding microtome (Leica SM 2000R) and subsequently emerged in cryoprotectant (35% ethylene glycol, 25% glycerol in PBS) and stored on –20°C. Hippocampal cultures were fixed after 5 weeks of cultivation period with 4% paraformaldehyde (PFA) in 0.1 M phosphate buffer (PB), pH 7.4 for 2 h. Cultures were rinsed three times with 0.1 M PBS for 10 min. The Millipore membrane with the fixed cultures was cut out and mounted onto a planar agar block. The cultures were sliced into 50-μm sections with a vibratome. Antigen retrieval was enhanced by heating the sections in 10 mM citrate buffer (1.8 mM citric acid, 8.2 mM trisodium citrate [pH 6.0]; 90°C; 35 min). For detection of α-synuclein phosphorylated at ser-129, we used a rabbit monoclonal pS129 antibody (Abcam, EP1536Y, 1:1,000) and followed the standard protocols provided with the VECTASTAIN Elite ABC Kit and the SG blue kit (Vector Laboratories, USA) (brain sections) or used an Alexa-fluorophore-conjugated secondary antibody (goat-anti-rabbit Alexa-568; Thermo Fisher, A11011) in a concentration of 1:250 (slice cultures). For Thioflavin S staining, sections were incubated for 5 min with freshly prepared filtered Thioflavin S solution (3% w/v Thioflavin S in Milli-Q $H_2O$) and washed 2× in 70% EtOH and 3× in Milli-Q $H_2O$ for 10 min each.

### Quantification of immunohistochemical stainings in slice cultures

For the quantification of the pS129-positive inclusions in HSCs, whole culture mosaic images were acquired on an Axioplan 2 imaging microscope (Plan Neofluar 10×/0.50 objective lens; Zeiss). Images were blinded, colour channels were split, background was subtracted (rolling ball radius 50 pixels), and the intensity threshold was manually adjusted. On each mosaic, the percentage of pS129-positive signal over the whole culture was calculated. Statistical analysis was done with GraphPad Prism (v.9), using unpaired two-tailed *t* test.

## The paper explained

### Problem

Neurodegenerative diseases are aggravating conditions with tremendous impact on the lives of affected people. Despite years of concerted scientific efforts, most of these conditions remain incurable. The initial step of causal therapies is often the identification of druggable targets that can interfere with disease progression. A transmembrane protein called lymphocyte-activation gene 3 (LAG3) whose function in the immune system is well established has recently been proposed to be a receptor of the disease-associated form of α-synuclein, a protein involved in α-synucleinopathies, amongst them Parkinson's disease.

### Results

Using an exhaustive array of *in vitro*, *ex vivo* and *in vivo* experiments, we have been unable to validate a role for LAG3 in α-synucleinopathies. We did not find evidence for LAG3 expression in neuronal cells of human or murine origin, and the interaction between LAG3 and α-synuclein fibrils appeared to be of limited specificity. LAG3 overexpression in human neural cells did not induce aggravation of α-synuclein pathology, and the genetic ablation of LAG3 in transgenic mice overexpressing human α-synuclein (A53T) did not lead to prolonged survival. Ultimately, the seeded induction of α-synuclein lesions in hippocampal slice cultures was unaffected by genetic depletion of LAG3.

### Impact

Our results question the relevance of LAG3 in the spreading of α-synucleinopathies, and thus, the quest for relevant targets to slow down, or completely abrogate, the pathogenesis of neurodegenerative diseases has to continue unabated. Although innovative approaches are needed to identify therapeutic candidates, emerging targets need to be rigorously validated, not only to maintain a stringent scientific record but also to moderate unjustified expectations from patients and other stakeholders.

## Data availability

The raw data underlying this study will be made available upon request. While the availability of human or murine samples is limited, we will be doing our utmost to share material provided strong scientific rationale. This study includes no data deposited in external repositories.

Expanded View for this article is available online.

## Acknowledgements

Rita Moos, Leyla Batkitar, Lidia Madrigal and Petra Schwarz provided technical assistance and help with mouse breeding, and Julie Domange, Marigona Imeri, Dezirae Schneider and Lisa Caflisch assisted with lab management (all at University of Zurich and University Hospital Zurich). We thank Tom Scheidt for initial discussions of microfluidic-based techniques and Georg Meisl for discussions on affinity. Institutional core funding by the University of Zurich and the University Hospital of Zurich to AA. Driver Grant 2017DRI17 of the Swiss Personalized Health Network (SPHN) and Distinguished Scientist Award of the NOMIS Foundation to AA. Grants of the European Research Council (ERC Prion2020, 670958 to AA and PhysProt to TPJK), EU Horizon 2020 research and innovation programme (ETN grant 674979-NANOTRANS) (MMS, TWH, NSM, TPJK). National Centre for Competence in Research (NCCR) RNA & Disease (51NF40-182880) and Swiss National Science Foundation Project Grant 310030_192650 to MP. EU/EFPIA/Innovative Medicines Initiative 2 Joint Undertaking (IMPRiND grant no 116060) to MJ. EDC is the recipient of an UZH Forschungskredit.

## Author contributions

AA, ME and EDC conceived the study and designed experiments. ME, MH-P, EDC, RGL and NB-V performed experiments shown in Fig 1. EDC, ET, ME, MH-P, PDR, AK and AG-G performed experiments shown in Fig 2. ME, MMS, EDC, TWH and NSM performed experiments shown in Fig 3. MH-P and ME performed experiments shown in Fig 4. TE, MB and MBac performed experiments shown in Fig 5. MH-P, EDC and NBV performed experiments shown in Fig EV1. EDC, NL and AK performed experiments shown in Fig EV2. ME performed experiments shown in Fig EV3. MH-P performed experiments shown in Fig EV4. LMH and TE performed experiments shown in Fig EV5. KL, RM, PJK, RR, MA and DH contributed reagents/materials. AA, MJ, MP, MRC, TPJK and SH supervised the study and provided advice. AA, ME, EDC and MH-P wrote the manuscript.

## Conflict of interest

AA is a member of the board of directors of Mabylon AG which has funded antibody-related work in the Aguzzi lab in the past. All other authors declare no competing interests.

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
