## [Review Process File · EMBO Molecular Medicine]

LAG3 is not expressed in human and murine neurons and does not modulate α -synucleinopathies

Marc Emmenegger, Elena De Cecco, Marian Hruska-Plochan, Timo Eninger, Matthias Schneider, Melanie Barth, Elena Tantardini, Pierre de Rossi, Mehtap Bacioglu, Rebekah Langston, Alice Kaganovich, Nora Bengoa-Vergniory, Andrés Gonzalez-Guerra, Merve Avar, Daniel Heinzer, Regina Reimann, Lisa Häsler, Therese Herling, Naunehal Matharu, Natalie Landeck, Kelvin Luk, Ronald Melki, Philipp Kahle, Simone Hornemann, Tuomas Knowles, Mark Cookson, Magdalini Polymenidou, Mathias Jucker, and Adriano Aguzzi

DOI: [10.15252/emmm.202114745](https://doi.org/10.15252/emmm.202114745)

Corresponding author: Adriano Aguzzi (adriano.aguzzi@usz.ch)

Review Timeline:

Transfer from Review Commons:	18th Jun 21
Editorial Decision:	22nd Jun 21
Revision Received:	25th Jun 21
Editorial Decision:	30th Jun 21
Revision Received:	5th Jul 21
Accepted:	6th Jul 21

Editor: Jingyi Hou

Transaction Report:

Review
COMMONS

This manuscript was transferred to EMBO Molecular Medicine following peer review at Review Commons.

Review #1

1. How much time do you estimate the authors will need to complete the suggested revisions:

Estimated time to Complete Revisions (Required)

(Decision Recommendation)

Less than 1 month

2. Evidence, reproducibility and clarity:

Evidence, reproducibility and clarity (Required)

Very high evidence and clarity. Excellent scientific rigor.

The findings are important and reported clearly. The experiments are conducted in a rigorous way by numerous participating laboratories.

3. Significance:

Significance (Required)

Very high significance, both from a molecular biology and clinical standpoints. This is an important manuscript that challenges the findings and conclusions of a prior high-profile paper in Science by Ma et al 2016, claiming that LAG3 is a receptor for aggregation-prone species of alpha-synuclein and that deletion of LAG3 results in reduced cell to cell propagation of alpha-synuclein aggregates.

The experiments in this paper are numerous and employ a variety of techniques. The overall conclusions are that LAG3 is not expressed by the relevant neurons and that LAG3 is not a receptor for alpha-synuclein fibrils (of different sizes). Therefore, the authors conclude that LAG3 is unlikely to play a role in the spread of alpha-synuclein pathology in Parkinson's disease and related disorders.

There are, however, some weaknesses. For example, the Introduction contains passages that are not written in a stringent way:

1. "Histologically, PD is characterized by α -synuclein aggregates known as Lewy Bodies in neurons of the substantia nigra," That is not a good description of PD neuropathology. Lewy pathology is present in numerous areas of the CNS and PNS, and is not restricted to the substantia nigra.
2. "Growing evidence suggests that α -synuclein fibrils spread from cell to cell". While alpha-synuclein pathology can spread from cell to cell, it is not known if the fibrils are

the species (alone or combined with other conformers) that cause the spreading of the pathology in a seeding fashion, or if smaller alpha-synuclein assemblies play that role. 3. "...by a "prionoid" process of templated conversion (Aguzzi, 2009; Aguzzi and Lakkaraju, 2016; Jucker and Walker, 2018; Kara, Marks and Aguzzi, 2018; Scheckel and Aguzzi, 2018; Uemura et al., 2020)." This sentence gives the impression that the corresponding author has led the field when it comes to alpha-synuclein's prionid properties. That is not really the case, and it would be appropriate to cite the literature in a more scholarly fashion that reflects how this part of the alpha-synuclein research field developed.

4. "Interrupting transmission of a-synuclein may slow down or abrogate the disease course." This is a bold statement and far from certain. While one might propose that this is the case, it is still just a hypothesis and the Introduction should reflect that.

****Referee Cross-commenting****

I concur with reviewers 2 and 3, and the new comment from reviewer 2. This paper should be published as soon as possible.

Review #2

1. How much time do you estimate the authors will need to complete the suggested revisions:

Estimated time to Complete Revisions (Required)

(Decision Recommendation)

Less than 1 month

2. Evidence, reproducibility and clarity:

Evidence, reproducibility and clarity (Required)

This study conclusively shows that LAG3 is not the receptor for a-synuclein that underlies the spread of synucleinopathic damage in various PD-related conditions. The paper is done extremely carefully and comprehensively. My only suggestion is to indicate the significance level in Figure 5a, as it may turn out that LAG3 is actually protective.

3. Significance:

Significance (Required)

This study is of extremely high significance - we need mechanisms to deal with spectacular results in the literature that should not have been published because they were unconvincing to begin with, but were published for various sociological/political reasons. Science won't progress if we don't find correction mechanisms for wrong conclusions.

****Referee Cross-commenting****

I agree with reviewers 1 and 3, especially with the suggestions made by reviewer 1, which should be instituted. I think we all concur that the paper should be published without new experiments. I believe testing a-synuclein propagation in vivo in LAG3 KO mice would be useful, but given the complete lack of replication of LAG3 expression in brain and of a-synuclein binding to LAG3, this is not necessary.

Review #3

1. How much time do you estimate the authors will need to complete the suggested revisions:

Estimated time to Complete Revisions (Required)

(Decision Recommendation)

Less than 1 month

2. Evidence, reproducibility and clarity:

Evidence, reproducibility and clarity (Required)

It was proposed that LAG3 is important in the treatment of PD and related disorders, because it functions as a receptor of pathogenic α -synuclein and the treatment with anti-LAG3 antibodies attenuated the spread of pathological α -synuclein and drastically lowered the aggregation in vitro (Mao et al, Science 2016).

In this study, authors characterized 8 antibodies to LAG3 and investigated the presence of LAG3 in cultured cell lines, NSC-derived neural cultures, or organ homogenates for the presence of human or murine LAG3. But it was not detected in any of the neuronal samples tested. In addition, single cell (sc) RNAseq yielded only minimal counts for the LAG3 transcript in neurons, astrocytes, and mixed glial cells, and single-nucleus (sn) RNAseq human brain dataset for LAG3 expression across different cell types confirmed no LAG3 signals for any of 34 identified cell clusters, including 13 clusters of excitatory and 11 subtypes of inhibitory neurons, oligodendrocytes, oligodendrocyte precursor cells, microglia, astrocytes, and endothelial cells.

Authors also analyzed the binding of LAG3 with α -synuclein in mouse and human model

systems, and concluded that the affinity of LAG3 for α -synuclein fibrils, if any, is micromolar or less.

Furthermore, authors studied the propagation of pre-formed fibrils (PFFs) of α -synuclein in neural stem cell (NSC)-derived neural cultures in the presence or absence of LAG3, and the impact of LAG3 on survival in ASYNA53T transgenic mice expressing wild-type LAG3 as well as hemizygous or homozygous deletions thereof. However, they were unable to see any significant role for LAG3 in these in vitro and in vivo models of α -synucleinopathies.

In this connection, the reviewer would like to ask one question: Have you conducted any experiments of the propagation of PFFs of α -synuclein in LAG3-KO mice ? If they did, what were the results ?

****Minor point****

In Page 10, I think it's a typo: ASYYN mice must be ASYN mice.

3. Significance:

Significance (Required)

These negative findings about the LAG in α -synucleinopathies shown in this manuscript do not provide any new insight into the mechanisms of α -synuclein propagation. However, it is clear that LAG3 is not expressed in neuronal cells and the binding of LAG3 to α -synuclein fibrils appears limited. Overexpression of LAG3 in cultured human neural cells did not cause any worsening of α -synuclein pathology ex vivo. The overall survival of A53T α -synuclein transgenic mice was unaffected by LAG3 depletion and the seeded induction of α -synuclein lesions in hippocampal slice cultures was unaffected by LAG3 knockout. These data shown in this manuscript are convincing and the information is very important in terms of correcting the direction of disease treatment and research.

****Referee Cross-commenting****

I agree with reviewers 1 and 2. This paper should be published as soon as possible.

Reviewer #1 (Evidence, reproducibility and clarity (Required)):

1. "Histologically, PD is characterized by α -synuclein aggregates known as Lewy Bodies in neurons of the substantia nigra," That is not a good description of PD neuropathology. Lewy pathology is present in numerous areas of the CNS and PNS and is not restricted to the substantia nigra.

We have added a more detailed account:

"Histologically, PD is characterized by α -synuclein inclusions known as Lewy Bodies whose accumulation is associated with neurodegeneration (Dickson, 2012; Mullin and Schapira, 2015; Corbillé et al., 2016). These inclusions affect the Substantia nigra and other mesencephalic regions as well as, in some cases, the amygdala and neocortex (Dickson, 2018)."

2. "Growing evidence suggests that α -synuclein fibrils spread from cell to cell". While alpha-synuclein pathology can spread from cell to cell, it is not known if the fibrils are the species (alone or combined with other conformers) that cause the spreading of the pathology in a seeding fashion, or if smaller alpha-synuclein assemblies play that role.

We have reformulated the sentence to credit the fact that we do not know which synuclein species is the one that is transmitted:

"Growing evidence suggests that α -synuclein aggregates spread from cell to cell (Volpicelli-Daley et al., 2011; Volpicelli-Daley, Luk and Lee, 2014)... "

3. "...by a "prionoid" process of templated conversion (Aguzzi, 2009; Aguzzi and Lakkaraju, 2016; Jucker and Walker, 2018; Kara, Marks and Aguzzi, 2018; Scheckel and Aguzzi, 2018; Uemura et al., 2020)." This sentence gives the impression that the corresponding author has led the field when it comes to alpha-synuclein's prionid properties. That is not really the case, and it would be appropriate to cite the literature in a more scholarly fashion that reflects how this part of the alpha-synuclein research field developed.

I cannot disagree, and in fact I suspect that the present paper may be my second and possibly last experimental contribution to the synuclein field! However, I do claim intellectual parenthood of the prionoid (not "prionid") concept, which I first expounded in a 2009 Nature paper. Anyway, we now provide a more balanced citation:

"...by a "prionoid" process of templated conversion (Aguzzi, 2009; Jucker and Walker, 2018; Kara, Marks and Aguzzi, 2018; Henderson, Trojanowski and Lee, 2019; Karpowicz, Trojanowski and Lee, 2019; Uemura et al., 2020; Kara et al., 2021)."

4. "Interrupting transmission of a-synuclein may slow down or abrogate the disease course." This is a bold statement and far from certain. While one might propose that this is the case, it is still just a hypothesis and the Introduction should reflect that.

We have rewritten the sentence in a more subdued manner:

"It is thought that interrupting transmission of a-synuclein may slow down or abrogate the disease course."

Reviewer #2 (Evidence, reproducibility and clarity (Required)):

This study conclusively shows that LAG3 is not the receptor for α -synuclein that underlies the spread of synucleinopathic damage in various PD-related conditions. The paper is done extremely carefully and comprehensively. My only suggestion is to indicate the significance level in Figure 5a, as it may turn out that LAG3 is actually protective.

We have added the significance level in Fig. 5A, in the legend: "The survivals of ASYN^{A53T} LAG3^{-/-}, LAG3^{+/-} and LAG3^{+/+} mice were similar (Mantel-Cox log-rank test, p-value = 0.165)."

****Referee Cross-commenting****

I agree with reviewers 1 and 3, especially with the suggestions made by reviewer 1, which should be instituted. I think we all concur that the paper should be published without new experiments. I believe testing α -synuclein propagation in vivo in LAG3 KO mice would be useful, but given the complete lack of replication of LAG3 expression in brain and of α -synuclein binding to LAG3, this is not necessary.

We considered running experiments in addition to those performed in vivo in ASYN^{A53T} transgenic mice (including LAG3 KO) and ex vivo in organotypic slices, the latter using pre-formed fibrils. However, the outcome of these experiments, along with the absence of LAG3 expression in neurons and its unclear binding, convinced us that the usage of further animals and reagents would be unwarranted.

Reviewer #3 (Evidence, reproducibility and clarity (Required)):

Have you conducted any experiments of the propagation of PFFs of α -synuclein in LAG3-KO mice ? If they did, what were the results ?

We did consider the possibility of replicating the experiments using PFFs in LAG3 KO mice. However, as stated above, we felt that our experiments – including the survival study in vivo in ASYN^{A53T} transgenic mice – were unambiguous. After critical consideration, we remained unconvinced that this additional experiment would change the weight of our evidence in a substantial manner that would justify the inoculation of other animals and the utilisation of more resources.

****Minor point****

In Page 10, I think it's a typo: ASYYN mice must be ASYN mice.

Thank you for pointing this out. We corrected it.

22nd Jun 2021

Thank you for the submission of your manuscript to EMBO Molecular Medicine. I have now had a chance to carefully read your manuscript and the point-by-point response to the concerns raised by Review Commons referees. I also discussed your work and your response to the referees' comments with the other members of our editorial team. We are satisfied with the modifications made and would like to invite a minor revision of your manuscript.

We noticed that in your point-by-point response, some of the referees' comments are missing. Please make sure that you include all the comments from referees, including the *Referee Cross-commenting*.

I look forward to receiving your revised manuscript soon.

Link Not Available

Kind regards,
Jingyi

Jingyi Hou
Editor
EMBO Molecular Medicine

*** IMPORTANT INFORMATION ***

We require:

3) A .docx formatted letter INCLUDING the reviewers' reports and your detailed point-by-point responses to their comments. As part of the EMBO Press transparent editorial process, the point-by-point response is part of the Review Process File (RPF), which will be published alongside your paper.

4) A complete author checklist, which you can download from our author guidelines (<https://www.embopress.org/page/journal/17574684/authorguide#submissionofrevisions>). Please insert information in the checklist that is also reflected in the manuscript. The completed author checklist will also be part of the RPF.

6) It is mandatory to include a 'Data Availability' section after the Materials and Methods. Before submitting your revision, primary datasets produced in this study need to be deposited in an appropriate public database, and the accession numbers and database listed under 'Data Availability'. Please remember to provide a reviewer password if the datasets are not yet public (see <https://www.embopress.org/page/journal/17574684/authorguide#dataavailability>).

7) For data quantification: please specify the name of the statistical test used to generate error bars and P values, the number (n) of independent experiments (specify technical or biological replicates) underlying each data point and the test used to calculate p-values in each figure legend. The figure legends should contain a basic description of n, P and the test applied. Graphs must include a description of the bars and the error bars (s.d., s.e.m.).

8) We would also encourage you to include the source data for figure panels that show essential data. Numerical data should be provided as individual .xls or .csv files (including a tab describing the data). For blots or microscopy, uncropped images should be submitted (using a zip archive if multiple images need to be supplied for one panel). Additional information on source data and instruction on how to label the files are available at .

.

- the medical issue you are addressing,

- the results obtained and

- their clinical impact.

13) Author contributions: the contribution of every author must be detailed in a separate section (before the acknowledgments).

14) A Conflict of Interest statement should be provided in the main text.

Rev_Com_number: RC-2021-00804

New_manu_number: EMM-2021-14745

Corr_author: Aguzzi

Title: LAG3 is not expressed in human and murine neurons and does not modulate α -synucleinopathies

Reviewer #1 (Evidence, reproducibility and clarity (Required)):

Very high evidence and clarity. Excellent scientific rigor.

The findings are important and reported clearly. The experiments are conducted in a rigorous way by numerous participating laboratories.

Reviewer #1 (Significance (Required)):

Very high significance, both from a molecular biology and clinical standpoints.

This is an important manuscript that challenges the findings and conclusions of a prior high-profile paper in Science by Ma et al 2016, claiming that LAG3 is a receptor for aggregation-prone species of alpha-synuclein and that deletion of LAG3 results in reduced cell to cell propagation of alpha-synuclein aggregates.

The experiments in this paper are numerous and employ a variety of techniques. The overall conclusions are that LAG3 is not expressed by the relevant neurons and that LAG3 is not a receptor for alpha-synuclein fibrils (of different sizes). Therefore, the authors conclude that LAG3 is unlikely to play a role in the spread of alpha-synuclein pathology in Parkinson's disease and related disorders.

There are, however, some weaknesses. For example, the Introduction contains passages that are not written in a stringent way:

1. "Histologically, PD is characterized by α -synuclein aggregates known as Lewy Bodies in neurons of the substantia nigra," That is not a good description of PD neuropathology. Lewy pathology is present in numerous areas of the CNS and PNS, and is not restricted to the substantia nigra.

We have added a more detailed account:

"Histologically, PD is characterized by α -synuclein inclusions known as Lewy Bodies whose accumulation is associated with neurodegeneration (Dickson, 2012; Mullin and Schapira, 2015; Corbillé et al., 2016). These inclusions affect the Substantia nigra and other mesencephalic regions as well as, in some cases, the amygdala and neocortex (Dickson, 2018)."

2. "Growing evidence suggests that α -synuclein fibrils spread from cell to cell". While alpha-synuclein pathology can spread from cell to cell, it is not known if the fibrils are the species (alone or combined with other conformers) that cause the spreading of the pathology in a seeding fashion, or if smaller alpha-synuclein assemblies play that role.

We have reformulated the sentence to credit the fact that we do not know which synuclein species is the one that is transmitted:

"Growing evidence suggests that α -synuclein aggregates spread from cell to cell (Volpicelli-Daley et al., 2011; Volpicelli-Daley, Luk and Lee, 2014)... "

3. "...by a "prionoid" process of templated conversion (Aguzzi, 2009; Aguzzi and Lakkaraju, 2016; Jucker and Walker, 2018; Kara, Marks and Aguzzi, 2018; Scheckel and Aguzzi, 2018; Uemura et al., 2020)." This sentence gives the impression that the corresponding author has led the field when it comes to alpha-synuclein's prionoid properties. That is not really the case, and it would be

appropriate to cite the literature in a more scholarly fashion that reflects how this part of the alpha-synuclein research field developed.

I cannot disagree, and in fact I suspect that the present paper may be my second and possibly last experimental contribution to the synuclein field! However, I do claim intellectual parenthood of the prionoid (not "prionid") concept, which I first expounded in a 2009 Nature paper. Anyway, we now provide a more balanced citation:

"...by a "prionoid" process of templated conversion (Aguzzi, 2009; Jucker and Walker, 2018; Kara, Marks and Aguzzi, 2018; Henderson, Trojanowski and Lee, 2019; Karpowicz, Trojanowski and Lee, 2019; Uemura et al., 2020; Kara et al., 2021)."

4. "Interrupting transmission of a-synuclein may slow down or abrogate the disease course." This is a bold statement and far from certain. While one might propose that this is the case, it is still just a hypothesis and the Introduction should reflect that.

We have rewritten the sentence in a more subdued manner:

"It is thought that interrupting transmission of a-synuclein may slow down or abrogate the disease course."

****Referee Cross-commenting****

I concur with reviewers 2 and 3, and the new comment from reviewer 2. This paper should be published as soon as possible.

Reviewer #2 (Evidence, reproducibility and clarity (Required)):

This study conclusively shows that LAG3 is not the receptor for a-synuclein that underlies the spread of synucleinopathic damage in various PD-related conditions. The paper is done extremely carefully and comprehensively. My only suggestion is to indicate the significance level in Figure 5a, as it may turn out that LAG3 is actually protective.

We have added the significance level in Fig. 5A, in the legend: "The survivals of ASYN^{A53T} LAG3^{-/-}, LAG3^{+/-} and LAG3^{+/+} mice were similar (Mantel-Cox log-rank test, p-value = 0.165)."

Reviewer #2 (Significance (Required)):

This study is of extremely high significance - we need mechanisms to deal with spectacular results in the literature that should not have been published because they were unconvincing to begin with, but were published for various sociological/political reasons. Science won't progress if we don't find correction mechanisms for wrong conclusions.

****Referee Cross-commenting****

I agree with reviewers 1 and 3, especially with the suggestions made by reviewer 1, which should be instituted. I think we all concur that the paper should be published without new experiments. I believe testing a-synuclein propagation in vivo in LAG3 KO mice would be useful, but given the

complete lack of replication of LAG3 expression in brain and of α -synuclein binding to LAG3, this is not necessary.

We considered running experiments in addition to those performed in vivo in ASYN^{A53T} transgenic mice (including LAG3 KO) and ex vivo in organotypic slices, the latter using pre-formed fibrils. However, the outcome of these experiments, along with the absence of LAG3 expression in neurons and its unclear binding, convinced us that the usage of further animals and reagents would be unwarranted.

Reviewer #3 (Evidence, reproducibility and clarity (Required)):

It was proposed that LAG3 is important in the treatment of PD and related disorders, because it functions as a receptor of pathogenic α -synuclein and the treatment with anti-LAG3 antibodies attenuated the spread of pathological α -synuclein and drastically lowered the aggregation in vitro (Mao et al, Science 2016).

In this study, authors characterized 8 antibodies to LAG3 and investigated the presence of LAG3 in cultured cell lines, NSC-derived neural cultures, or organ homogenates for the presence of human or murine LAG3. But it was not detected in any of the neuronal samples tested. In addition, single cell (sc) RNAseq yielded only minimal counts for the LAG3 transcript in neurons, astrocytes, and mixed glial cells, and single-nucleus (sn) RNAseq human brain dataset for LAG3 expression across different cell types confirmed no LAG3 signals for any of 34 identified cell clusters, including 13 clusters of excitatory and 11 subtypes of inhibitory neurons, oligodendrocytes, oligodendrocyte precursor cells, microglia, astrocytes, and endothelial cells.

Authors also analyzed the binding of LAG3 with α -synuclein in mouse and human model systems, and concluded that the affinity of LAG3 for α -synuclein fibrils, if any, is micromolar or less.

Furthermore, authors studied the propagation of pre-formed fibrils (PFFs) of α -synuclein in neural stem cell (NSC)-derived neural cultures in the presence or absence of LAG3, and the impact of LAG3 on survival in ASYNA53T transgenic mice expressing wild-type LAG3 as well as hemizygous or homozygous deletions thereof. However, they were unable to see any significant role for LAG3 in these in vitro and in vivo models of α -synucleinopathies.

In this connection, the reviewer would like to ask one question: Have you conducted any experiments of the propagation of PFFs of α -synuclein in LAG3-KO mice ? If they did, what were the results ?

We did consider the possibility of replicating the experiments using PFFs in LAG3 KO mice. However, as stated above, we felt that our experiments – including the survival study in vivo in ASYN^{A53T} transgenic mice – were unambiguous. After critical consideration, we remained unconvinced that this additional experiment would change the weight of our evidence in a substantial manner that would justify the inoculation of other animals and the utilisation of more resources.

****Minor point****

In Page 10, I think it's a typo: ASYYN mice must be ASYN mice.

Thank you for pointing this out. We corrected it.

Reviewer #3 (Significance (Required)):

These negative findings about the LAG in α -synucleinopathies shown in this manuscript do not provide any new insight into the mechanisms of α -synuclein propagation. However, it is clear that LAG3 is not expressed in neuronal cells and the binding of LAG3 to α -synuclein fibrils appears limited. Overexpression of LAG3 in cultured human neural cells did not cause any worsening of α -synuclein pathology ex vivo. The overall survival of A53T α -synuclein transgenic mice was unaffected by LAG3 depletion and the seeded induction of α -synuclein lesions in hippocampal slice cultures was unaffected by LAG3 knockout. These data shown in this manuscript are convincing and the information is very important in terms of correcting the direction of disease treatment and research.

****Referee Cross-commenting****

I agree with reviewers 1 and 2. This paper should be published as soon as possible.

30th Jun 2021 ,

Thank you for submitting your revised manuscript to EMBO Molecular Medicine, and thank you for addressing the editorial issues. I am pleased to inform you that we will be able to accept your manuscript pending the following amendments:

1. in the main manuscript file:

- please provide up to 5 keywords and incorporate them in the main text.
- please rename "Competing Interests" to "Conflicts of interest".
- Reference format: list all 10 co-authors of a paper before to add et al. in the reference list.
- 'FUNDING' needs to be merged with 'Acknowledgements'.
- In the Materials and Methods, include a statement that informed consent was obtained from all subjects.
- For animal work, the manuscript must include a statement in the Materials and Methods identifying the institutional and/or licensing committee approving the experiments. Housing conditions should be indicated.

2. Figures: Panel A can be removed for Figs EV3 and EV5 since there are no other panels. Please update the callouts accordingly.

3. Tables: The antibodies tables at the beginning of Materials and Methods need a title and callouts, and should be moved to the end of the manuscript.

4. Data Availability:

- Please rename "Data and material availability" into "Data availability", and place this section after the Materials & Methods section.
- Since this study does not generate large-scale datasets, please only include the following sentence in this section- "This study includes no data deposited in external repositories".

5. Please move "The paper explained" to the main manuscript file.

6. We would also encourage you to include the source data for figure panels that show essential data. Numerical data should be provided as individual .xls or .csv files (including a tab describing the data). For blots or microscopy, uncropped images should be submitted (using a zip archive if multiple images need to be supplied for one panel). Additional information on source data and instruction on how to label the files are available at

<https://www.embopress.org/page/journal/17574684/authorguide#sourcedata>

7. I only made a minor change in the synopsis text (see attached). I have also adjusted your synopsis image to the requested resolution (see attached).

Can you please let us know whether you are okay with the modified text and resized image, or if you would like to introduce future modifications?

Please note that this would be the final version, and synopsis text and image changes during

proofing are usually not allowed.

8. As part of the EMBO Publications transparent editorial process initiative (see our Editorial at <http://embomolmed.embopress.org/content/2/9/329>), EMBO Molecular Medicine will publish online a Review Process File (RPF) to accompany accepted manuscripts.

a. In the event of acceptance, this file will be published in conjunction with your paper and will include the anonymous referee reports, your point-by-point response and all pertinent correspondence relating to the manuscript. Let us know if you do not agree with this.

9. Our data editors have seen the manuscript, and they have made some comments and suggestions that need to be addressed (see attached). Please send back a revised version (in track change mode), as we will need to go through the changes.

I look forward to receiving your revised manuscript soon.

Kind regards,
Jingyi

Jingyi Hou
Editor
EMBO Molecular Medicine

*** Instructions to submit your revised manuscript ***

To submit your manuscript, please follow this link:

<https://embomolmed.msubmit.net/cgi-bin/main.plex>

1) a .docx formatted version of the manuscript text (including Figure legends and tables)

2) Separate figure files*

3) supplemental information as Expanded View and/or Appendix. Please carefully check the authors guidelines for formatting Expanded view and Appendix figures and tables at <https://www.embopress.org/page/journal/17574684/authorguide#expandedview>

4) a letter INCLUDING the reviewer's reports and your detailed responses to their comments (as Word file).

5) The paper explained: EMBO Molecular Medicine articles are accompanied by a summary of the articles to emphasize the major findings in the paper and their medical implications for the non-specialist reader. Please provide a draft summary of your article highlighting

6) For more information: There is space at the end of each article to list relevant web links for further consultation by our readers. Could you identify some relevant ones and provide such information as well? Some examples are patient associations, relevant databases, OMIM/proteins/genes links, author's websites, etc...

7) Author contributions: the contribution of every author must be detailed in a separate section.

8) EMBO Molecular Medicine now requires a complete author checklist (<https://www.embopress.org/page/journal/17574684/authorguide>) to be submitted with all revised manuscripts. Please use the checklist as guideline for the sort of information we need WITHIN the manuscript. The checklist should only be filled with page numbers where the information can be found. This is particularly important for animal reporting, antibody dilutions (missing) and exact values and n that should be indicated instead of a range.

9) Every published paper now includes a 'Synopsis' to further enhance discoverability. Synopses are displayed on the journal webpage and are freely accessible to all readers. They include a short stand first (maximum of 300 characters, including space) as well as 2-5 one sentence bullet points that summarise the paper. Please write the bullet points to summarise the key NEW findings. They should be designed to be complementary to the abstract - i.e. not repeat the same text. We encourage inclusion of key acronyms and quantitative information (maximum of 30 words / bullet point). Please use the passive voice. Please attach these in a separate file or send them by email, we will incorporate them accordingly.

You are also welcome to suggest a striking image or visual abstract to illustrate your article. If you do please provide a jpeg file 550 px-wide x 400-px high.

10) A Conflict of Interest statement should be provided in the main text

11) Please note that we now mandate that all corresponding authors list an ORCID digital identifier. This takes <90 seconds to complete. We encourage all authors to supply an ORCID identifier, which

will be linked to their name for unambiguous name identification.

Currently, our records indicate that the ORCID for your account is 0000-0002-0344-6708.

Link Not Available

12) The system will prompt you to fill in your funding and payment information. This will allow Wiley to send you a quote for the article processing charge (APC) in case of acceptance. This quote takes into account any reduction or fee waivers that you may be eligible for. Authors do not need to pay any fees before their manuscript is accepted and transferred to our publisher.

Photos 400-800 DPI

*Additional important information regarding figures and illustrations can be found at <https://bit.ly/EMBOPressFigurePreparationGuideline>

The system will prompt you to fill in your funding and payment information. This will allow Wiley to send you a quote for the article processing charge (APC) in case of acceptance. This quote takes into account any reduction or fee waivers that you may be eligible for. Authors do not need to pay any fees before their manuscript is accepted and transferred to our publisher.

1. in the main manuscript file:

- please provide up to 5 keywords and incorporate them in the main text.

Done.

- please rename "Competing Interests" to "Conflicts of interest".

Done.

- Reference format: list all 10 co-authors of a paper before to add et al. in the reference list.

Adjusted.

- 'FUNDING' needs to be merged with 'Acknowledgements'.

Done.

- In the Materials and Methods, include a statement that informed consent was obtained from all subjects.

Done.

- For animal work, the manuscript must include a statement in the Materials and Methods identifying the institutional and/or licensing committee approving the experiments. Housing conditions should be indicated.

Done.

2. Figures: Panel A can be removed for Figs EV3 and EV5 since there are no other panels. Please update the callouts accordingly.

Done.

3. Tables: The antibodies tables at the beginning of Materials and Methods need a title and callouts, and should be moved to the end of the manuscript.

Done.

4. Data Availability:

- Please rename "Data and material availability" into "Data availability", and place this section after the Materials & Methods section.

Done.

- Since this study does not generate large-scale datasets, please only include the following sentence in this section- "This study includes no data deposited in external repositories".

Done.

5. Please move "The paper explained" to the main manuscript file.

Done.

6. We would also encourage you to include the source data for figure panels that show essential data. Numerical data should be provided as individual .xls or .csv files (including a

tab describing the data). For blots or microscopy, uncropped images should be submitted (using a zip archive if multiple images need to be supplied for one panel). Additional information on source data and instruction on how to label the files are available at

<https://www.embopress.org/page/journal/17574684/authorguide#sourcedata>

We are eager to provide everything helpful to the community. We have now put together a zip file that includes all source files.

7. I only made a minor change in the synopsis text (see attached). I have also adjusted your synopsis image to the requested resolution (see attached).

Can you please let us know whether you are okay with the modified text and resized image, or if you would like to introduce future modifications?

We are fine with the modified text.

We adjusted the image slightly and provide both .ai as well as .png files.

Please note that this would be the final version, and synopsis text and image changes during proofing are usually not allowed.

8. As part of the EMBO Publications transparent editorial process initiative (see our Editorial at <http://embomolmed.embopress.org/content/2/9/329>), EMBO Molecular Medicine will publish online a Review Process File (RPF) to accompany accepted manuscripts.

a. In the event of acceptance, this file will be published in conjunction with your paper and will include the anonymous referee reports, your point-by-point response and all pertinent correspondence relating to the manuscript. Let us know if you do not agree with this.

We agree with this.

9. Our data editors have seen the manuscript, and they have made some comments and suggestions that need to be addressed (see attached). Please send back a revised version (in track change mode), as we will need to go through the changes.

We adjusted everything as per request. All changes are in track-change mode.

6th Jul 2021

We are pleased to inform you that your manuscript is accepted for publication and is now being sent to our publisher to be included in the next available issue of EMBO Molecular Medicine.

We would like to remind you that as part of the EMBO Publications transparent editorial process initiative, EMBO Molecular Medicine will publish a Review Process File online to accompany accepted manuscripts. If you do NOT want the file to be published or would like to exclude figures, please immediately inform the editorial office via e-mail.

Please read below for additional IMPORTANT information regarding your article, its publication and the production process.

Congratulations on your interesting work,
Jingyi

Jingyi Hou
Editor
EMBO Molecular Medicine

Follow us on Twitter @EmboMolMed
Sign up for eTOCs at embopress.org/alertsfeeds

*** ** IMPORTANT INFORMATION ** **

SPEED OF PUBLICATION

The journal aims for rapid publication of papers, using the advance online publication "Early View" to expedite the process: A properly copy-edited and formatted version will be published as "Early View" after the proofs have been corrected. Please help the Editors and publisher avoid delays by providing e-mail address(es), telephone and fax numbers at which author(s) can be contacted.

Should you be planning a Press Release on your article, please get in contact with embomolmed@wiley.com as early as possible, in order to coordinate publication and release dates.

LICENSE AND PAYMENT:

All articles published in EMBO Molecular Medicine are fully open access: immediately and freely available to read, download and share.

EMBO Molecular Medicine charges an article processing charge (APC) to cover the publication

costs. You, as the corresponding author for this manuscript, should have already received a quote with the article processing fee separately. Please let us know in case this quote has not been received.

Once your article is at Wiley for editorial production you will receive an email from Wiley's Author Services system, which will ask you to log in and will present you with the publication license form for completion. Within the same system the publication fee can be paid by credit card, an invoice, pro forma invoice or purchase order can be requested.

Payment of the publication charge and the signed Open Access Agreement form must be received before the article can be published online.

PROOFS

You will receive the proofs by e-mail approximately 2 weeks after all relevant files have been sent to our Production Office. Please return them within 48 hours and if there should be any problems, please contact the production office at embopressproduction@wiley.com.

Please inform us if there is likely to be any difficulty in reaching you at the above address at that time. Failure to meet our deadlines may result in a delay of publication.

All further communications concerning your paper proofs should quote reference number EMM-2021-14745-V3 and be directed to the production office at embopressproduction@wiley.com.

Thank you,

Jingyi Hou
Editor
EMBO Molecular Medicine

Corresponding Author Name: Adriano Aguzzi

Manuscript Number: EMM-2021-14745